# ONLINE CONTINUAL LEARNING FOR INTERACTIVE INSTRUCTION FOLLOWING AGENTS

**Byeonghwi Kim**[1,*]  **Minhyuk Seo**[1,*]   **Jonghyun Choi**[2,†]
[1]Yonsei University   [2]Seoul National University
{byeonghwikim,dbd0508}@yonsei.ac.kr, jonghyunchoi@snu.ac.kr

## ABSTRACT

In learning an embodied agent executing daily tasks via language directives, the literature largely assumes that the agent learns all training data at the beginning. We argue that such a learning scenario is less realistic since a robotic agent is supposed to learn the world continuously as it explores and perceives it. To take a step towards a more realistic embodied agent learning scenario, we propose two continual learning setups for embodied agents; learning new behaviors (Behavior Incremental Learning, Behavior-IL) and new environments (Environment Incremental Learning, Environment-IL) For the tasks, previous *'data prior'* based continual learning methods maintain logits for the past tasks. However, the stored information is often insufficiently learned information and requires task boundary information, which might not always be available. Here, we propose to update them based on confidence scores without task boundary information during training (*i.e.*, *task-free*) in a moving average fashion, named Confidence-Aware Moving Average (CAMA). In the proposed Behavior-IL and Environment-IL setups, our simple CAMA outperforms prior state of the art in our empirical validations by noticeable margins. The project page including codes is https://github.com/snumprlab/cl-alfred.

## 1 INTRODUCTION

Recent advances in computer vision, natural language processing, and embodied AI have led to various benchmarks for robotic agents, encompassing navigation (Savva et al., 2019; Deitke et al., 2020; Anderson et al., 2018; Krantz et al., 2020), object interaction (Zhu et al., 2017; Misra et al., 2017; Weihs et al., 2021; Ehsani et al., 2021), and interactive reasoning (Das et al., 2018; Gordon et al., 2018). To create more realistic agents, challenging benchmarks (Shridhar et al., 2020; Padmakumar et al., 2022) require all of these tasks to complete complex tasks based on language directives.

However, most embodied AI literature assumes that all training data are available from the outset but it may be unrealistic as agents may encounter novel behaviors or environments after deployment. To learn new behaviors and environments, continual learning might be necessary for post-deployment.

To learn new tasks, one may finetune the agents. But the finetuned agents would suffer from catastrophic forgetting that loses previously learned knowledge (McCloskey & Cohen, 1989; Ratcliff, 1990). To mitigate such forgetting, (Powers et al., 2022) introduced a continual reinforcement learning framework that incrementally updates agents for new tasks and evaluates their knowledge of current and past tasks. However, this operates in a simplified task setup of (Shridhar et al., 2020), excluding natural language understanding and object localization.

Taking a step forward to bring the instruction following task to real-world scenarios, we propose two continual learning scenarios for embodied agents: Behavior Incremental Learning (Behavior-IL) and Environment Incremental Learning (Environment-IL) as depicted in Figure 1. In Behavior-IL, the robot learns behaviors incrementally. For example, it may initially learn object movement and subsequently acquire the skill of object heating. In Environment-IL, instead of being limited to specific scenes such as bathrooms, the robot progressively learns to perform behaviors in diverse environments such as kitchens and bedrooms.

---

*Equal contribution. †Corresponding author. Most of the work is done while JC is with Yonsei University.

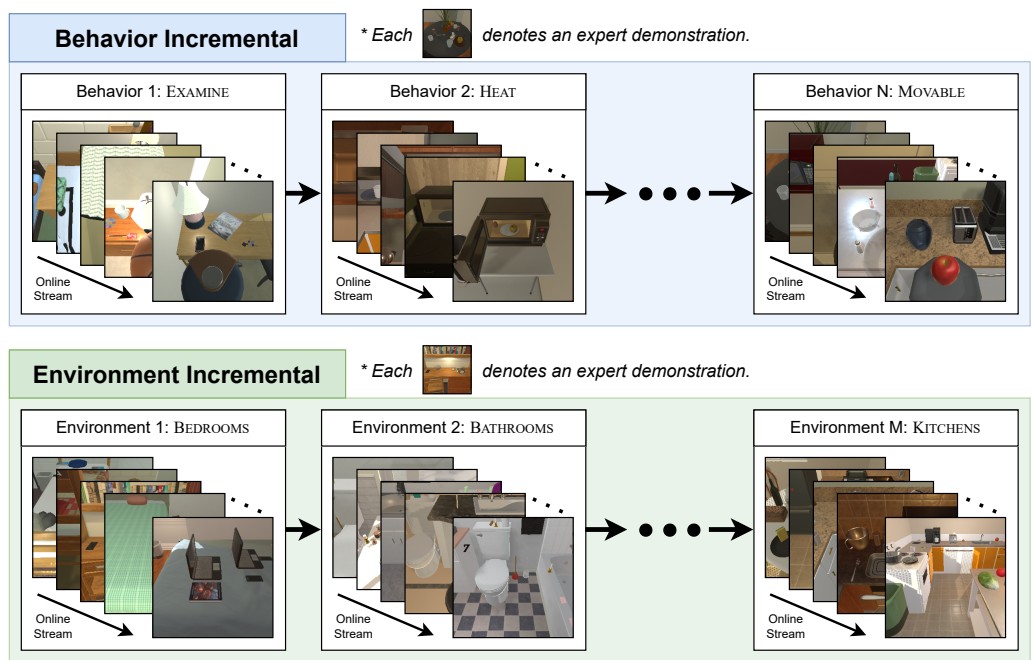

Figure 1: **Proposed two incremental learning setups.** In the 'Behavior Incremental' setup, the agent is expected to learn new behaviors while preserving previously learned knowledge. In the 'Environment Incremental' setup, the agent is expected to learn tasks in new environments with the preservation of previously learned knowledge. Note that each image in the figure denotes an expert demonstration (*i.e.*, a sequence of frames with natural language instructions followed by a corresponding sequence of actions and object class labels).

In the continual learning literature, significant progress (Mai et al., 2022; Biesialska et al., 2020) has been made in addressing continual learning by storing models learned in the previous task of extracting information about past data, requiring a substantial storage cost (Zhou et al., 2022a). To address this, Buzzega et al. (2020); Boschini et al. (2022a) propose to store logits of past models for knowledge distillation, reducing storage costs while maintaining learning efficacy. However, the stored logits may be the underfitted or insufficiently learned solution as the model has not sufficiently trained in the early stage of learning, hindering the effective use of prior knowledge. Moreover, such an update often exploits task boundary information that might not always be available, especially in the cases of streamed data without explicit task boundaries (Shanahan et al., 2021; Koh et al., 2023).

To develop continuously updating embodied agents, we propose to update logits by combining the previously stored logits and the newly obtained ones in the moving average, call *'Confidence-Aware Moving Average'* (CAMA). In particular, we dynamically determine the moving average coefficients based on the classification confidence scores inferred by the agents as indicators of the 'quality' of the newly obtained logits (*i.e.*, how much they contain accurate knowledge of the corresponding tasks), as empirically observed that high confidence tends to have high accuracy in Figure 3.

**Contributions.** We summarize our contributions as follows:
- We propose behavior incremental (Behavior-IL) and environment incremental (Environment-IL) setups for online continual learning for interactive instruction following agents.
- We propose Confidence-Aware Moving Average (CAMA) that dynamically determines coefficients for logit update to prevent logits from being outdated for effective knowledge distillation.
- Our proposed method outperforms comparable methods in most metrics with noticeable margins.

## 2 RELATED WORK

**Continual learning setups.** Continual Learning (CL) are typically categorized into two main scenarios: *offline* (Rebuffi et al., 2017; Kirkpatrick et al., 2017; Chaudhry et al., 2018; Wu et al., 2019) and *online* (Koh et al., 2022; Buzzega et al., 2020; Mai et al., 2022; Koh et al., 2023; Aljundi et al., 2019a), based on the frequency with which the model accesses task data. In the offline setup, data from the current task are used for training multiple times, but this often requires significant memory

capacity to store all task data (Hayes & Kanan, 2022). On the other hand, online CL involves processing individual or small batches of samples, each of which was used only once for training (Koh et al., 2022; Aljundi et al., 2019b). Considering memory constraints and the continuous arrival of limited data points over time in practical scenarios, we focus on the online CL setup.

**Task-free continual learning setups.** Approaches to continual learning can be categorized into task-free methods (Aljundi et al., 2019a; Koh et al., 2023; Ye & Bors, 2022; Koh et al., 2022) and task-aware methods (Kirkpatrick et al., 2017; Li & Hoiem, 2017; Wu et al., 2019; Boschini et al., 2022a) based on the use of task boundary information during training (Aljundi et al., 2018b). Several task-aware methods (Li & Hoiem, 2017; Wu et al., 2019) that exploit task boundary information during training distill the knowledge of past tasks from the previously learned model saved in memory. However, in real-world scenarios, it is often impractical to know the task identity of streamed input data (Aljundi et al., 2018b). Therefore, even if tasks are discretely defined (*e.g.*, blurry (Prabhu et al., 2020; Bang et al., 2021), disjoint (Parisi et al., 2019)), the task-free assumption indicates that no task-specific information or identifier is used during training.

**Knowledge distillation in online continual learning.** Continual learning has made significant progress, employing methods such as replay-based (Wu et al., 2019; Rolnick et al., 2019; Prabhu et al., 2020; Bang et al., 2021; Koh et al., 2022), distillation (Li & Hoiem, 2017; Buzzega et al., 2020; Koh et al., 2023), and regularization (Zenke et al., 2017; Kirkpatrick et al., 2017; Lesort et al., 2019). In particular, distillation-based approaches (Li & Hoiem, 2017; Koh et al., 2023; Boschini et al., 2022b) have been extensively investigated to leverage prior knowledge, but often require substantial memory and additional computation. Memory requirements make them unsuitable for settings with limited memory in edge devices (Zhou et al., 2022a).

To address these issues, (Buzzega et al., 2020; Michieli & Zanuttigh, 2021; Boschini et al., 2022a) propose using logits instead of storing previous models, saving memory and inference overhead. However, the method of storing logits in memory without updating may hinder the current model in distilling outdated past information from stored logits, since the stored logits may represent incomplete learning for past tasks. To tackle this, a recent approach X-DER (Boschini et al., 2022a) updates logits through weighted summation with logits maintained in memory and those from the current model, preventing the previously stored logit from becoming outdated, as the model updates.

However, Boschini et al. (2022a) requires task boundary information during the training process for logit update, making it unsuitable for setups where data arrive in a continuous stream without task boundaries. In contrast, our approach updates logits based on the agent's confidence without task boundaries, making it suitable for more general setups where we do not have such information.

**Lifelong learning for robotic agents.** Going beyond relatively straightforward task setups such as image classification, a substantial amount of literature has emerged to construct a more realistic agent capable of incremental learning of novel tasks (Lesort et al., 2020) in various aspects including learning strategies (*e.g.*, reinforcement learning (Khetarpal et al., 2018; Wołczyk et al., 2021; Xie & Finn, 2022), imitation learning (Mendez et al., 2018; Gao et al., 2021), *etc.*) and task formulations (*e.g.*, manipulation (Yang et al., 2022; Liu et al., 2023), walking (Zhou et al., 2022b), *etc.*). Typically, most of this research has focused mainly on relatively fine-grained manipulation tasks, while the navigation aspect (Krantz et al., 2020; Deitke et al., 2020) has received comparatively less attention.

Concurrently, there is recent literature that delves into the dual aspect of navigation and interaction (Powers et al., 2022; Wang et al., 2023) in 3D interactive environments (Kolve et al., 2017; Fan et al., 2022) to perform more demanding tasks. In this context, agents are required to become proficient in both navigating and interacting with task-related objects. Here, the tasks in our proposed continual learning setups are similar to (Powers et al., 2022) that simplifies the task setup of (Shridhar et al., 2020). While (Powers et al., 2022) excludes natural language understanding and object localization, we include them to train agents to complete the desired tasks in the challenging full-fledged interactive instruction following setup, along with navigation and object interaction.

We review more relevant literature and provide extended related work in Sec. A for space's sake.

## 3   CL-ALFRED: CONTINUAL LEARNING SETUPS FOR EMBODIED AGENTS

Continual learning enables agents to adapt to new behaviors and diverse environments after deployment, mitigating the risk of forgetting previously acquired knowledge. To foster active research

on mitigating catastrophic forgetting, recent literature (Powers et al., 2022) proposes a benchmark that continuously learns new types of household tasks, but lacks natural language understanding and object localization, potentially limiting the deployability of agents.

To comprehensively address the combined challenges of continuous learning of an agent with natural language understanding and object localization, we use full-fledged interactive instruction following tasks and propose two incremental learning setups, *Behavior Incremental Learning* (**Behavior-IL**) and *Environment Incremental Learning* (**Environment-IL**). We outline our task formulation and detail these incremental learning setups in the following sections.

## 3.1 TASK FORMULATION

As the ALFRED dataset (Shridhar et al., 2020) requires a comprehensive understanding of natural language and visual environments for intelligent agents, we build our continual benchmark on top of it. The agent is first spawned at a random location in an environment and then given natural language instructions, $l$, that describe how to accomplish a task. For each time step $t$, the agent takes as input visual observation $v_t$ and predicts an action $y_{a,t}$ and a mask $y_{m,t}$ of an object class $y_{c,t}$ if object interaction is required. Here, we learn a policy parameterized by $\theta$, $\pi_\theta : x \to y$, with input $x_t$, *i.e.*, $(v_t, l)$ and output $y_t$, *i.e.*, $(y_{a,t}, y_{m,t})$. The goal for the policy $\pi_\theta$ is to predict a sequence of actions and object masks to complete the task. Kindly refer to Shridhar et al. (2020) for more details.

Most previous methods (Singh et al., 2021; Pashevich et al., 2021; Min et al., 2022; Kim et al., 2023) for object localization utilize a two-stage approach, separating it into object class prediction and mask generation to enhance object localization. Since mask generation is handled by separate mask generators, however, continual updates of these networks are also required. Unfortunately, continuously updated instance segmentation models (Joseph et al., 2021; Cermelli et al., 2022) often noticeably underperform jointly trained models. Here, we only address class prediction, assuming the availability of object masks, leaving the continual updating of mask generators for future work.

## 3.2 CONTINUAL LEARNING SETUPS

There is significant progress in developing agents that can perform the desired tasks through language directives (Krantz et al., 2020; Shridhar et al., 2020; Padmakumar et al., 2022). It is often confronted with new behaviors and environments after being deployed and are required to learn them while maintaining previously learned knowledge. However, prior methods either presuppose the availability of pre-collected datasets or utilize simplified task configurations (Powers et al., 2022).

To address this limitation, we introduce two continual learning setups: 1) **Behavior Incremental Learning** (**Behavior-IL**) to incrementally learn *what* to do and 2) **Environment Incremental Learning** (**Environment-IL**) to incrementally learn *where* to do, as in Figure 1. In addition, we focus on online CL, which assumes a more realistic scenario where novel data are encountered in a streaming manner (Aljundi et al., 2019a; Koh et al., 2023; 2022) rather than assuming an offline CL in which novel data are provided in chunks of tasks (Wu et al., 2019; Saha et al., 2021). More details about the continual setup can be found in Sec. B.1.

### 3.2.1 BEHAVIOR INCREMENTAL LEARNING

Behaviors described by instructions may exhibit considerable diversity as novel behaviors may arise over time. To address this scenario, we propose the Behavior-IL setup that facilitates the agent's incremental learning of new behaviors while retaining the knowledge obtained from previous tasks.

Formally, for a set of behaviors, $\mathcal{T}$, an agent sequentially receives $N_j$ training episodes, $\{s_i^{\tau_j}\}_{i=1}^{N_j}$, for each type of behavior, $\tau_j \in \mathcal{T}$. When receiving the final episode (*i.e.*, $s_{N_j}^{\tau_j}$) for the current behavior type, $\tau_j$, the agent starts to sequentially receive episodes, $\{s_i^{\tau_{j+1}}\}_{i=1}^{N_{j+1}}$, for the next behavior type, $\tau_{j+1}$. The episode stream ends with the last training episode, $s_{N_{|\mathcal{T}|}}^{\tau_{|\mathcal{T}|}}$, of the last task type, $\tau_{|\mathcal{T}|}$.

Here, we adopt seven different types of behavior from (Shridhar et al., 2020): EXAMINE, PICK&PLACE, HEAT, COOL, CLEAN, PICK2&PLACE, and MOVABLE. To ensure the adaptability of agents and avoid favoring particular behavior sequences, we train and evaluate agents using multiple randomly ordered behavior sequences. Refer to Sec. B.2 for more details about the sequences.

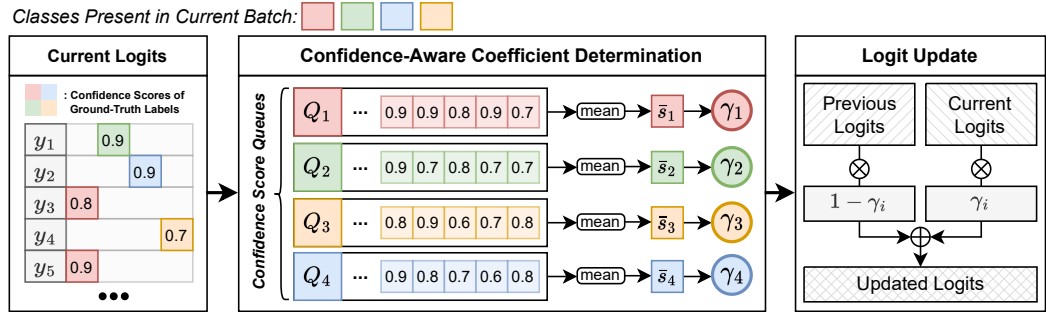

Figure 2: **Proposed Confidence-Aware Moving Average (CAMA).** 'Current Logits' denotes the model's logits obtained from the input samples from the current stream and episodic memory. 'Previous Logits' denotes logits stored in episodic memory before an update. $Q_i$ denotes a queue that stores ground truth confidence scores acquired from the current logits, $y_1, y_2, ...,$ for the $i^{th}$ class. To obtain $\gamma_i$, we maintain the recent $N$ confidence scores for the $i^{th}$ class and calculate the mean value of the scores followed by a clip function. Finally, we use all $\gamma_i$'s to class-wisely weight-sum previously stored logits (*i.e.*, 'Previous Logits') and newly obtained logits from the current stream (*i.e.*, 'Current Logits').

### 3.2.2 ENVIRONMENT INCREMENTAL LEARNING

The Environment-IL setup allows agents to learn the environment incrementally. In the real world, agents often need to perform actions not only in the environment in which they were initially trained but also in new and different environments that are presented. For example, the agent may first learn various actions in a kitchen and then subsequently learn the actions in a bathroom.

Formally, for a set of environments, $\mathcal{E}$, an agent sequentially receives $M_k$ training episodes, $\{s_i^{e_k}\}_{i=1}^{M_k}$, for each environment type, $e_k \in \mathcal{E}$. When receiving the final episode (*i.e.*, $s_{M_k}^{e_k}$) for the current environment type, $e_k$, the agent begins to sequentially receive episodes, $\{s_i^{e_{k+1}}\}_{i=1}^{M_{k+1}}$, for the next environment type, $e_{k+1}$. This is repeated until it reaches the last environment type, $e_{|\mathcal{E}|}$.

In this setup, we adopt four different types of environments supported by (Kolve et al., 2017): KITCHENS, LIVINGROOMS, BEDROOMS, and BATHROOMS. Like the Behavior-IL setup, we conduct training and evaluation using multiple sequences of randomly ordered environments. We also provide more details of the multiple environment sequences in Sec. B.3.

However, we observe an imbalance in the training and evaluation episodes between different types of environment (Shridhar et al., 2020), where a majority of them originate from a specific environment type in many instances. The imbalance can potentially lead to biased learning of a model towards the dominant (*i.e.*, majority) environment type (Chakraborty & Chakraborty, 2020; Zhao et al., 2021a). To address the issue, we balance them by simply subsampling the training and evaluation episodes for each environment to match the number of episodes across environment types equally.

## 4 APPROACH

To mitigate catastrophic forgetting, recent approaches (Li & Hoiem, 2017; Koh et al., 2023; Boschini et al., 2022b) use knowledge distillation with the trained model until past tasks, but this often entails significant memory (Zhou et al., 2022a) and computational overhead caused by additional model inference (Prabhu et al., 2023). Due to the limitations in memory and computation on edge devices (Lee et al., 2022), logit distillation methods (Buzzega et al., 2020; Boschini et al., 2022a) have been proposed as alternatives to those that store entire models for distillation (Li & Hoiem, 2017; Koh et al., 2023), offering improved memory and computation efficiency. Despite such improved efficiency, some of the logit distillation

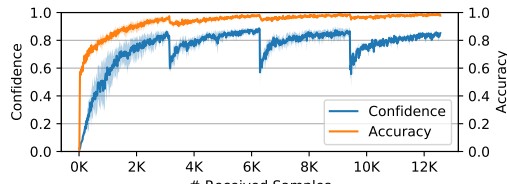

Figure 3: **Confidence and accuracy of logits used for logit update in CAMA.** 'Accuracy' denotes the mean of the frame-wise accuracies measured from the newly obtained logits (here, $z_a'$) in Equation 1. 'Confidence' denotes the dynamically determined coefficients (here, $\gamma_a$) for the update in Equation 2.

methods (Buzzega et al., 2020) often face an outdated logit problem, as the memory-stored logits are not updated to preserve information on previous tasks.

To address this issue, a recent approach (Boschini et al., 2022a) attempts to partially update logits stored in the past using the current model. It uses task boundary information (*e.g.*, input's task identity) during training, but it may not always be available, especially in *task-free* CL setups including our proposed ones. Towards a general logit-update framework devoid of such information, we update the stored logits using the agent's confidence scores indicating how the newly obtained logits for update contain accurate knowledge of the corresponding tasks, as observed in Figure 3.

### 4.1 CONFIDENCE-AWARE MOVING AVERAGE

As illustrated in Figure 2, the overall process of Confidence-Aware Moving Average (CAMA) can be summarized as follows: Initially, exploiting the model's confidence scores of ground-truth labels, we evaluate the extent to which the model has acquired proficiency in the current samples. Subsequently, during the computation of the updated logits based on the previous and current logits, we allocate a higher weight to the current logits when exhibiting higher confidence scores, and conversely, assign a higher weight to the previous logits demonstrating lower confidence scores. For better understanding, we outline the high-level flow of our CAMA in Algorithm 1 in the appendix.

Following the common practice (Kirkpatrick et al., 2017; Rolnick et al., 2019; Buzzega et al., 2020), we construct an input batch, $[x; x']$, by combining data from both the training data stream $(x, y_a, y_c) \sim \mathcal{D}$ and the episodic memory $(x', y'_a, y'_c, z'_{old,a}, z'_{old,c}) \sim \mathcal{M}$, where each $a \in \mathcal{A}$ and $c \in \mathcal{C}$ indicates an action and object class label from the action and object class sets, $\mathcal{A}$ and $\mathcal{C}$, present in the input batch, $[x; x']$. Here, $x$ represents the input (*i.e.*, images and language directives), $y_a$ and $y_c$ denote the corresponding action and object class labels, and $z'_{old,a}$ and $z'_{old,c}$ refers to the corresponding stored logits. $z_a, z_c, z'_a$, and $z'_c$ denote the current model's logits for the input batch.

To prevent the logits maintained in the episodic memory from becoming outdated, we obtain the updated logits, $z'_{new,a}$ and $z'_{new,c}$, by weighted-summing $z'_{old,a}$ and $z'_{old,c}$ with $z'_a$ and $z'_c$ using coefficient vectors, $\gamma_a$ and $\gamma_c$, using Hadamard product, denoted by $\odot$, as in Equation 1:

$$z'_{new,a} = (\mathbf{1} - \gamma_a) \odot z'_{old,a} + \gamma_a \odot z'_a, \quad z'_{new,c} = (\mathbf{1} - \gamma_c) \odot z'_{new,c} + \gamma_c \odot z'_c. \tag{1}$$

To obtain $\gamma_a$ and $\gamma_c$, we first maintain the most recent $N$ confidence scores for each action and object class label for $x$. Then, to approximate the agent's proficiency in learning tasks over time, we compute the average of the scores associated with each action ($i$) and object class ($j$) label, denoted by $\bar{s}_i^a$ and $\bar{s}_j^c$. We then set each element of $\gamma_a$ and $\gamma_c$, denoted by $\gamma_{a,i}$ and $\gamma_{c,j}$, to $\bar{s}_i^a$ and $\bar{s}_j^c$ followed by a CLIP function as in Equation 2:

$$\gamma_{a,i} = \alpha_a \text{CLIP} \left( \bar{s}_i^a - |\mathcal{A}|^{-1}, 0, 1 \right), \quad \gamma_{c,j} = \alpha_c \text{CLIP} \left( \bar{s}_j^c - |\mathcal{C}|^{-1}, 0, 1 \right), \tag{2}$$

where $\text{CLIP}(x, \min, \max)$ denotes the clip function that limits the value of $x$ from $\min$ to $\max$. Here, the constants $\alpha_a < 1$ and $\alpha_c < 1$ are introduced to prevent $\gamma_{a,i}$ and $\gamma_{c,j}$ from reaching a value of 1 as this indicates complete replacement of the prior logits with the current logits, which implies that the updated logits would entirely forget the previously learned knowledge. The inclusion of these constants ensures that some information from the past is retained and not entirely overridden by the current logits during the update process. In addition, we subtract $|\mathcal{A}|^{-1}$ and $|\mathcal{C}|^{-1}$ enhances the effective utilization of confidence scores in comparison to a 'random' selection, which would otherwise be realized by a uniform distribution (Koh et al., 2022).

### 4.2 MODEL TRAINING

Given expert demonstrations, $x$ as input, we train our agent, $\pi_\theta$, by minimizing the objective below:

$$\min_\theta \mathbb{E}_{(x,y) \sim \mathcal{D}}[\mathcal{L}(\pi_\theta(x), y)] + \mathbb{E}_{(x,y) \sim \mathcal{M}}[\mathcal{L}(\pi_\theta(x), y)] + \alpha \, \mathbb{E}_{(x,z) \sim \mathcal{M}}[||z - \pi_\theta(x)||_2^2], \tag{3}$$

where $y$ denotes the ground-truth labels corresponding to $x$ and $z$ the logits maintained in the episodic memory. We provide more details of the training loss, $\mathcal{L}$, in Sec. D.3 in the appendix.

## 5 EXPERIMENTS

**Evaluation metrics.** For evaluation of task completion ability, we follow the same evaluation protocol of (Shridhar et al., 2020). The primary metric is the success rate (SR) which measures the ratio of the succeeded episodes among the total ones. The second metric is the goal-condition success rate (GC) which measures the ratio of the satisfied goal conditions among the total ones. Furthermore, we evaluated all agents in two splits of environments: *seen* and *unseen* environments which are/are not used to train agents. We provide more details of the evaluation protocol in Sec. D.1.

To evaluate the last and intermediate performance of continual learning agents, we measure two variations of a metric, $A$: $A_{last}$ and $A_{avg}$. $A_{last}$ (here, $SR_{last}$ and $GC_{last}$) indicates the metric achieved by the agent that finishes its training for the final task. $A_{avg}$ (here, $SR_{avg}$ and $GC_{avg}$) indicates the average of the metrics achieved by the agents that finish their training for each incremental task.

All the models are trained sequentially over a sequence of behaviors (Behavior-IL) and environments (Environment-IL) and then evaluated over the behaviors and environments that the models have learned so far. For evaluation, we use episodes different from those used for training. The same trained models are evaluated in both *seen* and *unseen* environments. For *seen* and *unseen*, we denote by *seen* the evaluation with the episodes generated from scenes used in training, while we denote by *unseen* the evaluation with the episodes generated from scenes not used in training.

**Baselines.** We compare our CAMA with competitive prior arts in continual learning literature: CLIB (Koh et al., 2022), DER++ (Buzzega et al., 2020), ER (Rolnick et al., 2019), MIR (Aljundi et al., 2019a), EWC (Kirkpatrick et al., 2017), and X-DER (Boschini et al., 2022a). In addition, we also compare our CAMA with two models: 'Joint' and 'Finetuning'. 'Joint' denotes that the agent is trained with all task data jointly, which works as an upper bound. 'Finetuning' denotes that the agent is fine-tuned for the new tasks or scene types, which can serve as one of the trivial solutions for continual setups. We provide further explanation for each baseline in Sec. D.2. We further detail the model architecture and training used for the methods above in Sec. D.3 for space's sake.

**Implementation details.** It is a common practice in continual learning literature (Bang et al., 2021; Koh et al., 2022; 2023) to set the size of episodic memory to less than 5%. To align with previous works in continual learning, we set the size of the episodic memory to $M = 500$ for expert demonstrations, which is approximately $2.38\%$ of the training episodes in the ALFRED benchmark (Shridhar et al., 2020). For our CAMA, we empirically set $\alpha_a = 0.99$ and $\alpha_c = 0.99$. We provide more implementation details such as hyperparameters in Sec. D.4 for space's sake.

### 5.1 COMPARISON WITH STATE OF THE ART

We present the quantitative results of our CAMA in Table 1-2. As mentioned in Sec. 3.2, we train and evaluate our CAMA and baselines for three random seeds and report the results with their average and standard deviation to avoid favoring particular behavior and environment sequences. We provide quantitative analyses in various aspects as follows.

**Joint training vs. Finetuning.** Before investigating the effectiveness of our CAMA, we first investigate how challenging the proposed Behavior-IL and Environment-IL setups are. We observe significant performance drops in 'Finetuning' compared to 'Joint' with 51.0% and 47.5% relative drops. This implies that simply finetuning agents to novel behaviors and environments cannot effectively address the forgetting caused by distribution shifts between behaviors and environments.

**Ours vs. Regularization-based model.** We observe that our CAMA achieves better performance than the regularization-based approach (*i.e.*, EWC++) with noticeable margins in both seen and unseen environments for all metrics and setups, indicating that regularizing changes in importance parameters may not effectively prevent forgetting than distilling knowledge from updated logits.

**Ours vs. Rehearsal-based models.** We observe that our CAMA outperforms all rehearsal-based approaches (*i.e.*, ER, MIR, and CLIB) for all metrics in both seen and unseen environments in both Behavior-IL and Environment-IL setups. We believe that this implies that solely depending on sample replay amidst rapid data distribution shifts can result in insufficient task forgetting mitigation and hinder the agent's ability to adapt to novel tasks, ultimately impeding effective task completion.

**Ours vs. Distillation-based models.** We compare our CAMA with the distillation-based approaches (*i.e.*, DER and X-DER) to investigate the effectiveness of our logit-update approach. First,

| Model | Validation Seen | | | | | | | |
|---|---|---|---|---|---|---|---|---|
| | Behavior-IL | | | | Environment-IL | | | |
| | $SR_{last}$ ↑ | $GC_{last}$ ↑ | $SR_{avg}$ ↑ | $GC_{avg}$ ↑ | $SR_{last}$ ↑ | $GC_{last}$ ↑ | $SR_{avg}$ ↑ | $GC_{avg}$ ↑ |
| Finetuning | $9.51 \pm 1.09$ | $20.39 \pm 0.61$ | $17.07 \pm 0.86$ | $26.11 \pm 0.95$ | $8.72 \pm 2.12$ | $15.56 \pm 1.29$ | $16.25 \pm 3.95$ | $21.40 \pm 4.65$ |
| EWC++ | $20.37 \pm 5.19$ | $29.32 \pm 5.92$ | $22.21 \pm 4.34$ | $30.97 \pm 4.26$ | $26.79 \pm 2.24$ | $36.79 \pm 1.83$ | $31.01 \pm 2.76$ | $40.56 \pm 2.22$ |
| ER | $26.71 \pm 1.49$ | $36.59 \pm 1.36$ | $27.67 \pm 2.08$ | $36.20 \pm 1.96$ | $30.28 \pm 1.07$ | $39.15 \pm 0.83$ | $34.72 \pm 1.56$ | $44.00 \pm 1.52$ |
| MIR | $30.27 \pm 1.33$ | $40.14 \pm 2.00$ | $28.12 \pm 1.78$ | $36.76 \pm 1.73$ | $27.50 \pm 1.48$ | $36.31 \pm 1.43$ | $31.81 \pm 0.81$ | $40.94 \pm 0.95$ |
| CLIB | $23.85 \pm 2.02$ | $34.25 \pm 1.81$ | $23.94 \pm 2.36$ | $32.65 \pm 2.22$ | $25.47 \pm 1.42$ | $34.63 \pm 1.55$ | $32.51 \pm 2.40$ | $41.25 \pm 2.34$ |
| DER++ | $29.15 \pm 1.29$ | $39.39 \pm 1.16$ | $27.49 \pm 2.27$ | $36.10 \pm 1.92$ | $28.25 \pm 1.18$ | $36.18 \pm 0.60$ | $28.68 \pm 2.54$ | $38.01 \pm 3.16$ |
| X-DER | $28.76 \pm 1.25$ | $38.45 \pm 1.18$ | $27.21 \pm 2.53$ | $35.88 \pm 2.22$ | $28.30 \pm 0.87$ | $37.02 \pm 0.61$ | $29.32 \pm 2.36$ | $39.13 \pm 2.59$ |
| CAMA w/o D.C. | $30.54 \pm 1.27$ | $39.92 \pm 1.50$ | $\mathbf{29.89 \pm 2.32}$ | $38.16 \pm 2.71$ | $\mathbf{31.55 \pm 0.87}$ | $\mathbf{39.29 \pm 1.03}$ | $34.49 \pm 2.29$ | $43.28 \pm 2.17$ |
| **CAMA (Ours)** | $\mathbf{30.71 \pm 0.78}$ | $\mathbf{40.85 \pm 0.73}$ | $29.67 \pm 2.66$ | $\mathbf{38.17 \pm 2.34}$ | $29.48 \pm 0.27$ | $38.13 \pm 0.85$ | $\mathbf{35.09 \pm 1.92}$ | $\mathbf{44.02 \pm 2.21}$ |
| Joint | $60.47 \pm 0.33$ | $65.77 \pm 0.78$ | $-$ | $-$ | $56.25 \pm 0.89$ | $62.13 \pm 0.84$ | $-$ | $-$ |

Table 1: **Comparison with state-of-the-art methods (validation seen).** The highest value per metric is in **bold**. We report the averages and standard deviations of multiple runs for random seeds as in Sec. 3.2.

| Model | Validation Unseen | | | | | | | |
|---|---|---|---|---|---|---|---|---|
| | Behavior-IL | | | | Environment-IL | | | |
| | $SR_{last}$ ↑ | $GC_{last}$ ↑ | $SR_{avg}$ ↑ | $GC_{avg}$ ↑ | $SR_{last}$ ↑ | $GC_{last}$ ↑ | $SR_{avg}$ ↑ | $GC_{avg}$ ↑ |
| Finetuning | $1.18 \pm 1.09$ | $12.09 \pm 1.65$ | $3.03 \pm 1.29$ | $13.95 \pm 1.01$ | $2.01 \pm 0.86$ | $11.20 \pm 1.92$ | $2.90 \pm 2.16$ | $13.53 \pm 4.33$ |
| EWC++ | $8.50 \pm 2.15$ | $21.63 \pm 3.60$ | $8.33 \pm 1.33$ | $20.71 \pm 1.99$ | $11.61 \pm 1.29$ | $28.47 \pm 0.83$ | $12.37 \pm 1.30$ | $29.90 \pm 1.30$ |
| ER | $9.43 \pm 1.25$ | $24.22 \pm 1.54$ | $9.47 \pm 1.51$ | $22.79 \pm 1.39$ | $11.44 \pm 1.36$ | $29.11 \pm 0.96$ | $14.25 \pm 1.47$ | $31.98 \pm 1.64$ |
| MIR | $11.01 \pm 1.16$ | $25.31 \pm 1.14$ | $10.88 \pm 1.51$ | $24.20 \pm 1.45$ | $12.01 \pm 0.61$ | $29.67 \pm 0.47$ | $12.58 \pm 0.80$ | $29.11 \pm 1.26$ |
| CLIB | $8.26 \pm 1.03$ | $22.00 \pm 1.31$ | $8.56 \pm 0.66$ | $21.03 \pm 1.15$ | $10.46 \pm 1.18$ | $27.40 \pm 0.78$ | $11.95 \pm 1.51$ | $29.93 \pm 2.05$ |
| DER++ | $13.16 \pm 5.56$ | $28.70 \pm 5.63$ | $10.60 \pm 4.04$ | $24.94 \pm 2.69$ | $10.29 \pm 1.05$ | $26.90 \pm 1.31$ | $10.25 \pm 1.72$ | $26.83 \pm 1.97$ |
| X-DER | $12.59 \pm 1.92$ | $28.10 \pm 2.05$ | $12.04 \pm 1.56$ | $25.50 \pm 1.48$ | $10.75 \pm 1.15$ | $28.37 \pm 1.05$ | $11.14 \pm 1.38$ | $28.56 \pm 1.44$ |
| CAMA w/o D.C. | $\mathbf{14.06 \pm 1.20}$ | $28.33 \pm 1.58$ | $12.52 \pm 1.46$ | $26.02 \pm 1.38$ | $13.57 \pm 1.25$ | $29.54 \pm 1.41$ | $12.78 \pm 0.57$ | $29.76 \pm 0.84$ |
| **CAMA (Ours)** | $13.64 \pm 0.94$ | $\mathbf{28.75 \pm 0.92}$ | $\mathbf{14.19 \pm 1.38}$ | $\mathbf{27.30 \pm 1.38}$ | $\mathbf{14.60 \pm 0.43}$ | $\mathbf{30.99 \pm 0.75}$ | $\mathbf{15.67 \pm 0.77}$ | $\mathbf{33.40 \pm 1.45}$ |
| Joint | $24.60 \pm 0.96$ | $38.24 \pm 1.55$ | $-$ | $-$ | $19.73 \pm 2.31$ | $39.02 \pm 0.53$ | $-$ | $-$ |

Table 2: **Comparison with state-of-the-art methods (validation unseen).** The highest value per metric is in **bold**. We report averages and standard deviations of multiple runs for random seeds as in Sec. 3.2.

we observe noticeable performance drops in DER, which does not update logits, compared to our CAMA for all metrics in seen and unseen environments in both Behavior-IL and Environment-IL setups, which highlights the importance of updating logits to prevent them from being outdated.

In addition, we observe that our CAMA outperforms X-DER, which partially updates logits only for novel classes, with noticeable margins for all metrics and environments in both the Behavior-IL and Environment-IL setups, highlighting the efficacy of our CAMA. We note that while X-DER updates logits based on task boundary information during training, our CAMA does not assume the availability of such information (*i.e.*, task-free), which highlights the generality of our CAMA.

## 5.2 THE EFFECTIVENESS OF DYNAMICALLY DETERMINED COEFFICIENTS

We investigate the effect of dynamically determined coefficients of our CAMA by fixing them with a constant value and provide the results ('CAMA' vs. 'CAMA w/o D.C.' in Table 1-2). 'CAMA w/o D.C.' assumes that the agent is *always* 100% confident in what it learns. Consequently, we directly set $\gamma_a$ and $\gamma_c$ to $\alpha_a$ and $\alpha_c$ by omitting the process of dynamically determined coefficients.

We observe that the ablation of dynamically determined coefficients consistently yields performance drops in all metrics in seen and unseen environments in both Behavior-IL and Environment-IL setups, indicating the importance of the process of finding such coefficients. This could be attributed to the fact that while logit updating with a constant coefficient helps mitigate the obsolescence of the logits to some extent, it also combines them with logits from the current model that lacks sufficient training for novel tasks, particularly during the initial phase of learning these tasks. Consequently, this can lead to performance degradation due to incomplete knowledge of these new tasks.

## 5.3 QUALITATIVE ANALYSIS

We provide qualitative examples of our CAMA in each Behavior-IL and Environment-IL setup by comparison with the naïve (*i.e.*, Finetuning) and prior best-performing (DER++) methods. For space's sake, we provide the qualitative example in the Environment-IL setup in Sec. D.5.

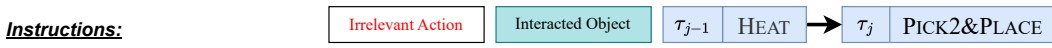

**Instructions:**

Turn around to your right move forward then turn left, head to the coffee maker. Pick up the mug in front of the coffee maker on the counter. Turn to your left head to the microwave above the stove. Open the microwave then put in and out the mug then close the microwave. Turn to your right and head to the coffee maker. Put the mug on the coffee maker.

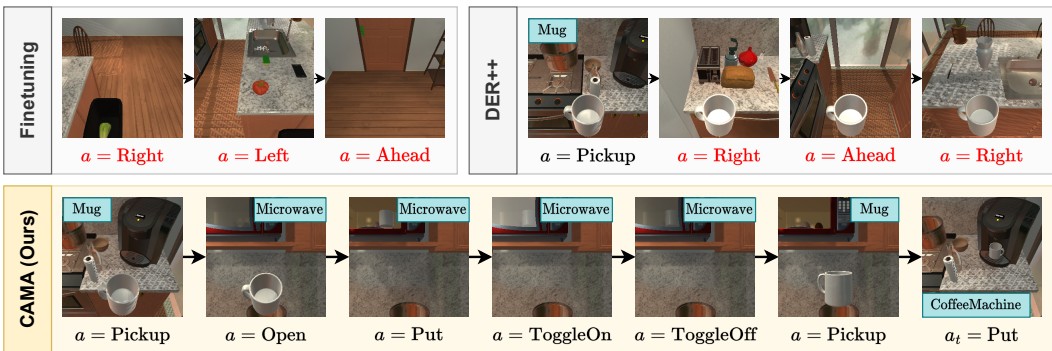

Figure 4: **Qualitative analysis of the proposed method (Behavior-IL).** The agent, having already acquired knowledge of the behavior $\tau_{j-1} = $ HEAT, proceeds to learn the new behavior $\tau_j = $ PICK2&PLACE. Subsequently, we evaluate the agent's ability of the prior behavior $\tau_{j-1}$ to determine if any forgetting has occurred. Irrelevant Action denotes an action that results in incorrect navigation. 'Finetuning' fails to find a target object, 'Mug,' and eventually fails at the task. DER++ succeeds in navigating to and picking up the mug but fails to reach a microwave above the agent, also leading to task failure. On the contrary, our CAMA further succeeds in reaching the microwave, heating the mug, and putting it back on the coffee machine, leading to task success.

**The Behavior-IL setup.** In Figure 4, the agent is evaluated for the previous behavior, $\tau_{j-1} = $ HEAT while learning the current behavior, $\tau_j = $ PICK2&PLACE. Here, the agent is required to heat a mug and put it on the coffee machine. The agent can sequentially complete the task by 1) picking up a mug, 2) heating it using a microwave, and 3) putting it on the coffee machine.

'Finetuning' first explores the environment to find a mug. However, it fails to recognize the mug and therefore keeps wandering in the environment, eventually leading to task failure. Meanwhile, 'DER++' succeeds in finding the mug and picking it up but forgets how to reach a microwave above the agent to heat an object. The agent also keeps wandering in the environment and eventually, it fails at the task. In contrast, our CAMA succeeds in navigating to and picking up the mug. After grabbing the mug, our agent finds, reaches the microwave above, and successfully heats the mug. Finally, our agent then put the heated mug on the coffee machine, as described in the instructions, which implies that our CAMA enables the agent to maintain the knowledge of the previous behaviors.

## 6 CONCLUSION

We propose two continual learning setups that learn new behaviors (Behavior Incremental Learning, Behavior-IL) and environments (Environment Incremental Learning, Environment-IL) continually. Prior methods employ the storage of model logits from previous tasks but they are updated either only once or upon obtaining new logits, potentially resulting in learning with outdated data or utilizing logits from a model that has incompletely learned the new tasks.

To effectively update the logits, we propose Confidence-Aware Moving Average (CAMA), a simple yet effective approach that dynamically determines moving average coefficients based on the agent's confidence scores. We observe that the CAMA outperforms all prior arts by noticeable margins.

**Limitation and future work.** While the disjoint setup operates under the assumption that tasks in streaming data are non-overlapping (Parisi et al., 2019), posing a stringent test for catastrophic forgetting, such non-overlapping scenarios might not always be the case in real-world scenarios. To address this aspect, extending our proposed setups to feature overlapped tasks in streaming data, such as blurry setups (Prabhu et al., 2020; Bang et al., 2021) or Gaussian scheduled regimes (Shanahan et al., 2021; Wang et al., 2022), represents a promising avenue for future research.

## ETHICS STATEMENT

This work introduces continual learning setups for interactive instruction following agents and a logit-update approach to enhance the effectiveness of knowledge distillation. While the authors do not aim for this, the increasing adoption of deep learning models in real-world contexts with streaming data could potentially raise concerns such as privacy and model robustness. There is a possibility of these deployed deep learning models inadvertently introducing biases or discrimination, as unresolved issues like model bias persist within deep learning. We are committed to implementing all feasible precautions to avert such consequences, as they are unequivocally contrary to our intentions.

## REPRODUCIBILITY STATEMENT

We take reproducibility in deep learning very seriously and highlight some of the contents of the manuscript that might help to reproduce our work. We release the data splits of the proposed benchmarks in Sec. 3, our implementation of the proposed method in Sec. 4, and the baselines used in our experiments in Sec. 5 in https://github.com/snumprlab/cl-alfred.

## ACKNOWLEDGMENT

This work was partly supported by the NRF grant (No.2022R1A2C4002300, 15%) and IITP grants (No.2020-0-01361 (10%, Yonsei AI), No.2021-0-01343 (5%, SNU AI), No.2022-0-00077 (10%), No.2022-0-00113 (20%), No.2022-0-00959 (15%), No.2022-0-00871 (15%), No.2021-0-02068 (5%, AI Innov. Hub), No.2022-0-00951 (5%)) funded by the Korea government (MSIT).

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

APPENDIX

## A  EXTENDED RELATED WORK

**Continual learning setup.**  We can categorize continual learning setups into Task Incremental (task-IL) and Class Incremental (class-IL) depending on whether task ID is given during inference. In task-IL, task-ID is provided during inference (Aljundi et al., 2018a; Lopez-Paz & Ranzato, 2017; Hossain et al., 2022), while in class-IL, task ID is not given (Koh et al., 2022; Buzzega et al., 2020; Koh et al., 2023; Bang et al., 2021). As task-ID is not provided during inference in the real world, class-IL is closer to the real setup and is more challenging because it requires classification across all classes. We focused on class-IL to deal with more realistic situations.

**Continual learning methods.**  As the need for continual learning is increasingly highlighted across various fields, researchers have proposed a wide array of continual learning methods to prevent catastrophic forgetting. (1) Replay-Based Methods (Prabhu et al., 2020; Koh et al., 2022; Bang et al., 2021; Wu et al., 2019; Rolnick et al., 2019) store some stream data in episodic memory and replaying memory data during the future learning process to prevent forgetting about previous tasks. (2) Distillation-Based Methods (Buzzega et al., 2020; Koh et al., 2023; Li & Hoiem, 2017) retain knowledge about past data by storing logits (Buzzega et al., 2020) or models (Koh et al., 2023; Li & Hoiem, 2017) to distill knowledge. (3) Regularization-Based Method (Zenke et al., 2017; Kirkpatrick et al., 2017; Lesort et al., 2019) prevents the overwriting of important parameters by imposing a penalty on changes to these crucial parameters.

Recently, continual learning has also been actively investigated in more challenging task setups such as video domains (Zhao et al., 2021b; Villa et al., 2022; Park et al., 2021). The focus of their work is generally on classification problems such as action recognition in a video by observing full video frames as given. In contrast, rather than receiving predetermined frames at once, the next observations (frames) of agents are determined by the actions that the agents take, which then require the agents to plan subsequent actions to complete tasks based on the next observations.

**Embodied AI.**  Embodied AI (EAI) has garnered substantial attention, and notable advancements have been made in various tasks (Anderson et al., 2018; Krantz et al., 2020; Jain et al., 2019; Savva et al., 2019; Deitke et al., 2020; Weihs et al., 2021; Ehsani et al., 2021; Shridhar et al., 2020; Padmakumar et al., 2022; Gao et al., 2022). For instance, visual navigation tasks necessitate that the agent use visual observation to reach designated locations (Savva et al., 2019) or objects (Savva et al., 2019; Deitke et al., 2020). Meanwhile, vision-language navigation (VLN) (Anderson et al., 2018; Jain et al., 2019; Krantz et al., 2020) augments visual observation with natural language descriptions, enabling the agent to plan a sequence of actions based on a comprehensive understanding of multiple modalities to successfully reach the target locations.

Furthermore, the scope of EAI tasks has expanded through the inclusion of object interaction. (Weihs et al., 2021) necessitates the agent to relocate objects to their original state by manipulating them, while (Ehsani et al., 2021) requires the agent to move objects to designated locations using six degrees of freedom (6-DoF) manipulation. (Shridhar et al., 2020) presents natural language descriptions, which the agent must comprehend to plan a sequence of actions and utilize predictive 2D object classes to locate objects for interaction. Meanwhile, (Padmakumar et al., 2022; Gao et al., 2022) provide natural language dialogues, in which the agent must engage in reasoning to determine the appropriate course of action and complete the tasks at hand.

However, the agents evaluated in those benchmarks are typically trained using pre-existing datasets. Given that the data collection process can be both expensive and time-consuming, it may not always be feasible to pre-collect the requisite dataset, implying the need for continual learning for the agents.

## B  ADDITIONAL CL-ALFRED BENCHMARK DETAILS

### B.1  CONTINUAL LEARNING SETUPS

In alignment with the common practice in the prior arts (Aljundi et al., 2019a; Buzzega et al., 2020; Shim et al., 2021), we assume that both setups follow an *online* and *disjoint* paradigm. In an *online* setup, individual samples (here, expert demonstrations) are presented sequentially rather than being

available simultaneously. While a portion of the data is retained in episodic memory, the streaming data is accessible for learning only once. In a *disjoint* setup, each task (here, each behavior and environment type) contains distinct and unrelated information from the others. In this setup, as the agent embarks on learning new tasks, it does not receive any samples from previously encountered tasks. Importantly, we do not rely on predefined task boundaries for training and evaluation.

## B.2 BEHAVIOR INCREMENTAL LEARNING

In the ALFRED benchmark, episodes comprise seven distinct behavior types: EXAMINE, HEAT, PICK&PLACE, COOL, CLEAN, PICK2&PLACE, and MOVABLE. Each behavior type presents distinct goal conditions that the agents must fulfill by learning how to achieve them. In the Behavior-IL setup, we employ the original validation split of the ALFRED benchmark for validation and five randomly ordered sequences of behavior types for training as follows.

1. EXAMINE → HEAT → PICK2&PLACE → COOL → PICK&PLACE → CLEAN → MOVABLE
2. PICK&PLACE → PICK2&PLACE → CLEAN → HEAT → EXAMINE → MOVABLE → COOL
3. PICK&PLACE → EXAMINE → MOVABLE → CLEAN → PICK2&PLACE → COOL → HEAT
4. MOVABLE → PICK2&PLACE → EXAMINE → PICK&PLACE → HEAT → COOL → CLEAN
5. CLEAN → PICK&PLACE → MOVABLE → HEAT → COOL → PICK2&PLACE → EXAMINE

## B.3 ENVIRONMENT INCREMENTAL LEARNING

Episodes in the ALFRED benchmark are generated from four distinct environment types supported by AI2-THOR (Kolve et al., 2017): KITCHENS, LIVINGROOMS, BEDROOMS, and BATHROOMS. Each environment type has 30 variations of the environment type that enable agents to learn behaviors in diverse room layouts and visual appearances. Similar to the Behavior-IL setup, for training, Environment-IL also employs five randomly ordered sequences of environment types as follows.

1. BEDROOMS → BATHROOMS → LIVINGROOMS → KITCHENS
2. BATHROOMS → BEDROOMS → KITCHENS → LIVINGROOMS
3. BEDROOMS → LIVINGROOMS → BATHROOMS → KITCHENS
4. BEDROOMS → BATHROOMS → KITCHENS → LIVINGROOMS
5. BATHROOMS → KITCHENS → BEDROOMS → LIVINGROOMS

As discussed in Sec. 3.2.2, we observe an imbalance in the 'train' and 'validation' splits of the original ALFRED across the environment types: for each KITCHENS, LIVINGROOMS, BEDROOMS, and BATHROOMS, $11,056, 3,456, 3,370$, and $3,141$ episodes for the 'train' split and $432, 129, 106$, and $153$ for the 'validation' *seen* split, and $468, 146, 120$, and $87$ for the 'validation' *unseen* split.

To balance them, we subsample the train and validation episodes per environment type as follows. For the 'train' split, we subsample $3,141$ episodes, leading to $12,564$ episodes in total. For the 'validation' *seen* split, we subsample $106$ episodes, leading to $424$ episodes in total. Finally, for the 'validation' *unseen* split, we subsample $87$ episodes, leading to $348$ episodes in total.

## C EXTENDED APPROACH

As described in Sec. 4.1, the high-level procedure of our CAMA is described in Algorithm 1. When new samples, denoted as $x$ (representing expert demonstrations), are received, we retrieve samples, denoted by $x'$, from episodic memory. Subsequently, we obtain the respective logits, denoted by $z$ and $z'$, from the model, denoted by $\pi_\theta$. With these logits, we compute the gradient of the joint loss, which combines cross entropy and knowledge distillation, to update the model parameters $\theta$.

After updating $\theta$, we maintain a record of the $N$ most recent confidence scores in separate queues, denoted by $Q$, for each action and object class for $x$. Once these recent scores are maintained, we dynamically calculate the coefficients, $\gamma_a$ and $\gamma_c$, to weight-sum the previous and current logits, denoted by $z'old$ and $z'$, leading to the updated logits, denoted by $z'new$, which are then stored in episodic memory. For more details on the $\gamma_a$ and $\gamma_c$ calculations, kindly refer to Sec. 4.1.

---

**Algorithm 1** Pseudo code for CAMA

---

**Input** Model $\pi_\theta$ parameterized by $\theta$, Memory $\mathcal{M}$, Training data stream $\mathcal{D}$, Learning rate $\mu$, Confidence score queues $Q$, Scalar $\beta$, $N$, Appeared action set $\mathcal{A}$, Appeared object class set $\mathcal{C}$

**for** $(x, y) \in \mathcal{D}$ **do**  // Samples arrive from the stream
    **Sample** $(x', y', z'_{\text{old}}) \leftarrow \text{RandomSample}(\mathcal{M})$  // Acquire triplets from the memory
    $z, z' \leftarrow f_\theta([x; x'])$  // Obtain logits from the model
    $\mathcal{L} = \text{CrossEntropyLoss}([z; z'], [y; y']) + \beta \|z' - z'_{\text{old}}\|_2^2$  // Compute the total loss
    $\theta \leftarrow \theta - \mu \cdot \nabla_\theta \mathcal{L}$
    $Q \leftarrow \text{MaintainRecentConfidences}(Q, N, z)$  // Maintain $N$ recent confidence scores
    $\gamma = \text{CalculateAdaptiveRatio}(\mathcal{A}, \mathcal{C}, Q)$
    $z'_{\text{new}} = \gamma z'_{\text{old}} + (1 - \gamma) z'$  // Update old logits by CAMA (Sec. 4.1)
    **Update** $\mathcal{M}(x', y', z'_{\text{old}}) \leftarrow \mathcal{M}(x', y', z'_{\text{new}})$  // Update logits for samples retrieved from memory
    $\mathcal{M} \leftarrow \text{ReservoirSampler}(\mathcal{M}, (x, y, z))$  // Update Memory
**end for**

---

## D   EXTENDED EXPERIMENT RESULTS

### D.1   EVALUATION METRICS

For training and evaluation, the ALFRED dataset (Shridhar et al., 2020) consists of three splits; 'train,' 'validation,' and 'test.' Agents are trained with the 'train' split and validate their approaches in the 'validation' split with the ground-truth information of the tasks in those splits. The agents are then evaluated in the 'validation' and 'test' split, but they do not have any access to the ground-truth information of the tasks. As the ground-truth labels of the 'test' split are not publicly available, we evaluate our agent and baselines and report the results in the validation split.

The validation split also consists of two folds: *seen* and *unseen* environments in which agents are/are not trained. *Seen* environments allow evaluating the task completion ability of agents in the same visual domain as training environments. *Unseen* environments further allow evaluating agents' task completion ability in *different* visual domains from training environments, which is considered more challenging than *seen* environments.

For the evaluation of task completion ability, the primary metric is the success rate (SR) which measures the ratio of completed tasks, indicating the task completion ability of the agent. Another metric is the goal-condition success rate (GC) which measures the ratio of satisfied goal conditions, indicating the partial task completion ability of the agent. We evaluate all agents' performance in terms of SR and GC in both *seen* and *unseen* environments and the main metric is *unseen* SR.

(Shridhar et al., 2020) also penalizes SR and GC with path-length-weighted (PLW) scores proportional to trajectory lengths. Considering the model (Kim et al., 2021) used in our experiments lags significantly behind human performance in terms of task completion ability (*i.e.*, *unseen* success rates), however, we focus mainly on task completion ability and leave the examination of efficiency aspects for future investigations.

### D.2   BASELINES

CLIB (Koh et al., 2022) is a method that maintains an optimal episodic memory based on the importance of each sample. DER++ (Buzzega et al., 2020) aims to distill information about past data by storing not only images and labels but also logits, comparing them with the logits of the current model. ER (Rolnick et al., 2019) constructs the training batch with half of the current task stream and the other half of the data in memory, to remember past data while learning about a new task. MIR (Aljundi et al., 2019a) uses samples that received the most interference from previous learning, rather than randomly retrieving from memory when composing the training batch. EWC++ (Kirkpatrick et al., 2017) prevents forgetting from parameter overwriting by penalizing changes of important parameters. X-DER (Boschini et al., 2022a) rewrites logits for the portions corresponding to the classes of past tasks to incorporate newly acquired experience information about past tasks while learning new tasks. Following the prior methods (Singh et al., 2021; Pashevich et al., 2021; Nguyen et al., 2021) that keep visual encoders frozen, our agent's visual encoder also remains frozen, and thus we omit the contrastive learning part (Chen et al., 2020) in X-DER.

## D.3 MODEL ARCHITECTURE AND TRAINING

For model architecture, we adopt a recently proposed learning-based agent (Kim et al., 2021) that perceives the surrounding views and predicts a sequence of actions and object masks using factorized branches (Singh et al., 2021). Following the common practice of prior arts (Shridhar et al., 2020; Singh et al., 2021; Pashevich et al., 2021; Nguyen et al., 2021; Kim et al., 2021), we train our agent with imitation learning, specifically behavior cloning. We detail the architecture and training below.

**Model Architecture.** The architecture of our CAMA and the baselines is based on a recent approach (Singh et al., 2021) that uses separate modules for effective action prediction and object localization to better address different semantic understandings from each other. Specifically, let $y_t = f(x_t)$ be the agent that maps the input $x_t = (v_t, l, y_{a,t-1})$ to the output $y_t = (y_{a,t}, y_{c,t})$. For the input, each $v_t$ and $l$ denotes the RGB images (*i.e.*, surrounding views) and step-by-step instructions. For the output, each $y_a$ and $y_c$ denotes the action and object class to be interacted with.

As mentioned above, the agent $f$ is comprised of two separate modules: the action prediction module $a_t = f_{action}(v_t, l, y_{a,t-1})$ and the object localization module $c_t = f_{class}(v_t, l, y_{a,t-1})$. Both modules encode the instructions $l$ with a self-attention-based Bi-LSTM network, resulting in the attended language feature, $\hat{l}$. To capture the correspondence between visual and textual information, we conduct point-wise convolution for $v_t$ by filters generated from $\hat{l}$, resulting in the attended visual feature $\hat{v}_t$. The decoder of each module updates its hidden state based on the attended visual and textual features, $\hat{v}_t$ and $\hat{l}$, with the previous action $y_{a,t-1}$, resulting in $h_t^a$ and $h_t^c$ for the action prediction module and the object localization module. Finally, fully connected layers in $f_{action}$ take as input $\hat{v}_t$, $\hat{l}$, $y_{a,t-1}$, and $h_t^a$ and predict the current action $y_{a,t}$. Similarly, fully connected layers in $f_{class}$ take as input $h_t^c$ and predict the current object class $y_{c,t}$ with which to interact.

For more details of the modules, kindly refer to (Singh et al., 2021).

**Training.** Following (Shridhar et al., 2020), we adopt imitation learning for training, specifically behavior cloning, that mimics an expert's behaviors in a teacher-forcing manner. Formally, let **a** and **a**$^*$ be a sequence of predicted actions and the corresponding ground-truth actions. Similarly, let **c** and **c**$^*$ be a sequence of predicted object classes to be interacted with and the corresponding labels. Then each loss of the action and object class prediction is obtained by a cross-entropy loss as below:

$$\mathcal{L}_{action}(\mathbf{a}, \mathbf{a}^*) = -\sum_{t=1}^{T} a_t^* \log a_t, \quad \mathcal{L}_{class}(\mathbf{c}, \mathbf{c}^*) = -\sum_{t=1}^{T} \mathbb{1}[a_t^* = \text{interaction}] \cdot c_t^* \log c_t, \quad (4)$$

where $T$ denotes the length of an episode that the agent conducts and $\mathbb{1}[a_t^* = \text{interaction}]$ is an indicator function that activates when an action $a_t^*$ is an object interaction action.

In addition, (Ma et al., 2019) adopts progress monitoring. Formally, let **p** and **p**$^*$ be a sequence of predicted progress values and the corresponding ground-truth progress values. Then the progress loss is obtained by a mean square error (MSE) loss as below:

$$\mathcal{L}_{progress}(\mathbf{p}, \mathbf{p}^*) = \frac{1}{T} \sum_{t=1}^{T} (p_t - p_t^*)^2. \quad (5)$$

Using them, the agent jointly minimizes the joint loss as follows:

$$\mathcal{L}(y, y^*) = \lambda_a \mathcal{L}_{action}(y_a, y_a^*) + \lambda_c \mathcal{L}_{class}(y_c, y_c^*) + \lambda_p \mathcal{L}_{progress}(y_p, y_p^*), \quad (6)$$

where $y$ indicates the output of a model including an action sequence, $y_a$, a class sequence, $y_c$, and a progress value sequence, $y_p$, for an auxiliary task. $y_a^*$, $y_c^*$, and $y_p^*$ denote the corresponding ground-truth labels. The loss terms are summed by the balancing coefficients $\lambda_a$, $\lambda_c$, and $\lambda_p$.

## D.4 IMPLEMENTATION DETAILS

For visual observation, inspired by (Nguyen et al., 2021; Kim et al., 2021; Bhambri et al., 2023), we allow the agent to perceive surrounding views (in this case, 5 views from the front, left, right, up, and down directions). For language instructions, the agents receive step-by-step instructions that describe how to accomplish the goal in detail.

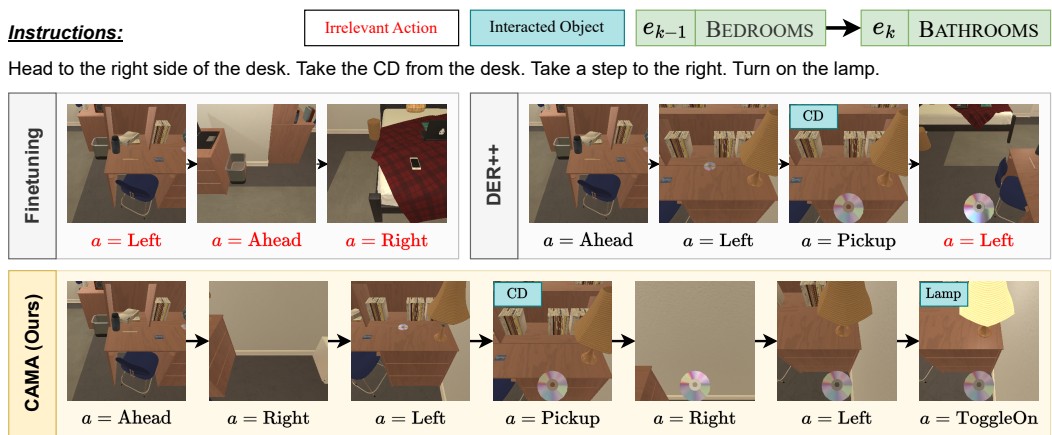

Figure 5: **Qualitative analysis of the proposed method (Environment-IL).** The agent that has already acquired knowledge of the environment $e_{k-1}$ = BEDROOMS proceeds to learn the new environment $e_k$ = BATHROOMS. We then assess the agent's capability in the prior environment $e_{k-1}$ to determine whether any forgetting has occurred. Irrelevant Action denotes an action that results in incorrect navigation. 'Finetuning' fails to find a target object, 'CD,' eventually leading to task failure. DER++ can navigate to and pick up the CD, but fails to turn on the lamp. On the contrary, our CAMA can also turn on the lamp and complete the task.

For the training loss described in Section D.3, we set the balancing coefficients $\lambda_a = 1.0$, $\lambda_c = 1.0$, and $\lambda_p = 1.0$ for our CAMA and the baselines. Following (Singh et al., 2021), we augment visual observations by adopting two strategies: AutoAugment (Cubuk et al., 2019) and RGB-channel swapping. For computational efficiency, we cache 6 types of augmented episodes per episode and choose one of them whenever we augment it.

To update the parameters of our CAMA and the baselines, we use the Adam optimizer with an initial learning rate of $0.001$ and a batch size of $32$ per streamed sample. We utilize the ExponentalLR (Li & Arora, 2019) and ResetLR (Loshchilov & Hutter, 2016) schedulers with $\gamma = 0.95$ and $m = 10$ for our CAMA and the baselines except CLIB with $\gamma = 0.9999$.

## D.5 QUALITATIVE ANALYSIS

We provide a qualitative example of our CAMA in the Environment-IL setup by comparison with the naïve (*i.e.*, Finetuning) and prior best-performing (DER++) methods.

**The Environment-IL setup.** In Figure 5, the agent is evaluated for the previous environment, $e_{k-1}$ = BEDROOMS, while learning the current environment, $e_k$ = BATHROOMS. The agent is required to examine a CD under the light of a lamp. To complete the task, the agent needs to 1) pick up a CD and 2) turn on a lamp while holding the CD.

Similarly in the Behavior-IL setup, 'Finetuning' cannot find a CD and therefore navigates to other objects irrelevant to the task, which eventually leads to task failure. On the other hand, 'DER++' successfully picks up a CD and reaches the lamp in the close vicinity. However, the agent forgets to turn on the lamp and therefore starts to navigate to other irrelevant objects, which also leads to task failure. In contrast, our CAMA can pick up the CD and navigate to the lamp as 'DER++' does. Our agent then turns on the light to examine the held CD and succeeds in the task.

## D.6 IMBALANCED SCENARIOS IN THE ENVIRONMENT-IL SETUP

To explore a data-imbalance scenario (Liu et al., 2022; He et al., 2023), we remove the subsampling process in Sec. 3.2.2 for the Environment-IL setup and construct an imbalanced dataset that we name the imbalanced Environment-IL setup. In the imbalanced Environment-IL setup, we compare our CAMA with the baselines and summarize the result in Table 3.

We observe that even with such an imbalance, CAMA still outperforms the baselines by noticeable margins, highlighting the efficacy of the proposed approach. In particular, we observe significant improvements of $SR_{avg}$ and $GC_{avg}$ in both valid seen and unseen splits, implying that CAMA achieves promising performance in both partial and full task completion.

| Model | Imbalanced Environment-IL | | | | | | | |
| --- | --- | --- | --- | --- | --- | --- | --- | --- |
| | Valid Seen | | | | Valid Unseen | | | |
| | $SR_{last}$ ↑ | $GC_{last}$ ↑ | $SR_{avg}$ ↑ | $GC_{avg}$ ↑ | $SR_{last}$ ↑ | $GC_{last}$ ↑ | $SR_{avg}$ ↑ | $GC_{avg}$ ↑ |
| EWC | $27.58 \pm 2.03$ | $38.58 \pm 2.40$ | $35.44 \pm 1.42$ | $45.14 \pm 1.58$ | $9.76 \pm 0.70$ | $22.94 \pm 0.61$ | $12.32 \pm 1.03$ | $28.06 \pm 1.31$ |
| ER | $32.22 \pm 1.98$ | $43.03 \pm 1.93$ | $38.87 \pm 0.29$ | $49.06 \pm 0.83$ | $11.80 \pm 0.86$ | $26.43 \pm 0.84$ | $13.43 \pm 0.61$ | $30.75 \pm 0.52$ |
| MIR | $27.58 \pm 2.03$ | $38.58 \pm 2.40$ | $35.44 \pm 1.42$ | $45.14 \pm 1.58$ | $9.76 \pm 0.70$ | $22.94 \pm 0.61$ | $12.32 \pm 1.03$ | $28.06 \pm 1.31$ |
| CLIB | $24.70 \pm 2.20$ | $35.21 \pm 2.78$ | $35.32 \pm 1.77$ | $44.55 \pm 1.78$ | $8.24 \pm 1.14$ | $21.96 \pm 1.11$ | $10.97 \pm 1.50$ | $28.03 \pm 1.11$ |
| DER++ | $31.54 \pm 1.42$ | $42.99 \pm 1.61$ | $33.01 \pm 1.97$ | $43.89 \pm 2.21$ | $12.20 \pm 0.69$ | $27.28 \pm 1.23$ | $11.90 \pm 1.46$ | $28.33 \pm 1.74$ |
| X-DER | $31.10 \pm 1.10$ | $42.83 \pm 1.13$ | $32.96 \pm 1.70$ | $43.81 \pm 1.64$ | $12.93 \pm 0.51$ | $27.44 \pm 0.97$ | $12.49 \pm 0.95$ | $29.07 \pm 1.30$ |
| CAMA w/o D.C. | $\mathbf{35.40 \pm 1.34}$ | $\mathbf{46.55 \pm 1.40}$ | $\mathbf{39.54 \pm 1.30}$ | $\mathbf{49.27 \pm 1.33}$ | $14.19 \pm 1.22$ | $\mathbf{29.11 \pm 1.42}$ | $16.02 \pm 1.20$ | $33.16 \pm 1.25$ |
| CAMA (Ours) | $32.32 \pm 1.62$ | $43.72 \pm 1.81$ | $38.67 \pm 2.07$ | $49.09 \pm 1.90$ | $\mathbf{14.53 \pm 0.40}$ | $28.75 \pm 0.92$ | $\mathbf{17.24 \pm 1.78}$ | $\mathbf{33.36 \pm 1.47}$ |

Table 3: **Comparison with state-of-the-art methods in the imbalanced Environment-IL setup.** The highest value per metric is in **bold**. We report the means and standard errors of multiple runs for random seeds.

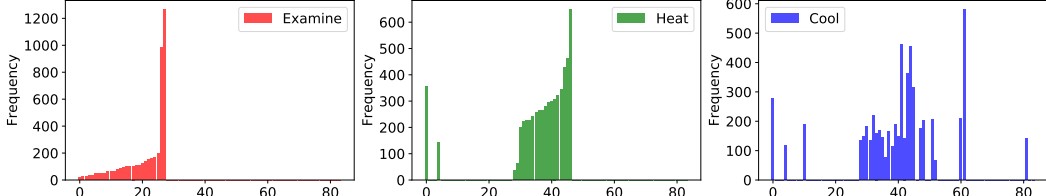

Figure 6: **The frequency of objects used for each behavior in the Behavior-IL setup.** Each x-axis and y-axis denotes the index of an object and the object's frequency appearing in the corresponding behavior. While the behaviors, HEAT and COOL, have several shared objects (*e.g.*, apples, tomatoes, *etc.*) during task completion, the behavior, EXAMINE, rarely have them with HEAT and COOL.

# E DISCUSSION

## E.1 CONFIDENCE SCORES AS A GOOD INDICATOR OF NEW LOGITS' QUALITY

We use the averaged class-wise confidence score as an indicator to estimate how much new logits are informative. This is because using the confidence scores for the ground truths, which we use for logit update, allows us to estimate how well the model has learned respective classes.

For example, If the model $p$ predicts $p(i) = 1$ for the class $i$, it implies that the model has learned the class $i$ well, *i.e.*, it may contain ample information about the class $i$. Conversely, if the model predicts $p(i) = 0$, it implies that the model has learned the class $i$ poorly, *i.e.*, it may contain little information about the class $i$ (Koh et al., 2023).

We can use the most *single* recent confidence score as the 'indicator,' but such a single confidence score could be noisy during training for various reasons such as the degree of augmentation and the difficulty of a sample. To alleviate this issue, we use the $N$ recent confidence scores as an indicator of the quality of the new logits.

## E.2 DEPENDENCY OF PERFORMANCE ON A TASK ORDER IN INCREMENTAL SETUPS

We agree that the performance improvements seem relatively marginal, possibly due to the large standard error of the means. We believe this is because, in an incremental setup in embodied tasks, some tasks may share relatively many action and object classes, while others may share fewer classes. Previously learning such shared action and object classes may help better learning the current task (*i.e.*, forward transfer) and this implies that a model's performance may depend on the order of the tasks (*i.e.*, how much the model learns the shared action and object classes ahead).

For example, while learning to cool an object, learning some actions (*e.g.*, opening/closing a fridge) and object classes (*e.g.*, apples, tomatoes, *etc.*) may help next learn to heat an object as such actions and object classes can also be used for heating (*e.g.*, heat an 'apple,' a 'tomato,' *etc.* by 'opening/closing' a microwave). We empirically observe that for the behavior, 'Heat,' our agent achieves 3.70% Valid Unseen SR after learning the behavior, 'Cool,' while it achieves zero Valid Unseen SR after learning the behavior, 'Examine,' which does have fewer shared object classes as illustrated in Figure 6, implying the dependency of performance on a task order.

