# OpenReview forum: "Online Continual Learning for Interactive Instruction Following Agents"
_ICLR.cc/2024/Conference — ICLR 2024 poster_

### Official Review · Reviewer_q27B · 2023-10-16

**Soundness:** 3 good
**Presentation:** 2 fair
**Contribution:** 2 fair
**Rating:** 6
**Confidence:** 2

**Summary:**

This paper first introduced two new settings to coninual learning embodied AI: behavior incremental learning and environmental incremental learning. The paper then introduces a method to tackle both, in a task boundariless setting, called Confidence-Aware Moving Average. The method is employed in Alfworld, and the task at each step is to predict action and object class.

**Strengths:**

- The continual learning setting is a relevant topic to be explored in the embodied AI field
- the writing is clear and the method is well motivated and easy to understand
- the number of baselines compared is very comprehensive
- the method is simple yet effective

**Weaknesses:**

- conceptually, the task boundariless scenario in embodied instruction following AI is very limited, since the user will know when to switch to a new task
- introduction to a new hyperparameter for each of the prediction ($\alpha$'s)
- another baseline I could think of is to use the loss as the weight to dynamically change the old logits. What are some intuition on why confidence is better than loss to use to dynamically weight?

**Questions:**

- equation (1), the $\gamma$  in front of $z$ is it missing the subscript $a$ and $c$? or is it a fixed value?
- for equation (2), how are the confidence scores calculated, mathematically
- I am not too familiar with the benchmark, but when is the object class used when making decisions? if you need to heat a mug, does your agent also need to recognize the object in front of it is a mug too?

---

> ### Author Response · Authors · 2023-11-22
> **Answers to the questions of Reviewer q27B (1/2)**
>
> We thank reviewer q27B for the constructive feedback and the encouraging remarks on a well-motivated and effective approach, clear presentation, and comprehensive baselines.
> We address your concerns as follows.
>
> > Conceptually, the task boundariless scenario in embodied instruction following AI is very limited, since the user will know when to switch to a new task.
>
> $\to$ First, we guess the ‘task boundariless’ you refer to is the ‘task free’ setup (in the second paragraph of Sec.4 and the ‘Ours vs. Distillation-based models.’ paragraph in Sec.5.1, and we will add more details about task-free setup in Sec.2 of the revision). We respectfully argue that it may not be trivial for the user to know when to switch to a new task. For example, when the data is injected in a Gaussian scheduled *continuous* scenario [1], the data distribution imperceptibly changes over time as there are no explicit task boundaries, and even the user cannot tell where the task boundary is. To address a large set of scenarios including the ones mentioned above, we devise a *task-free* method.
>
> [1] Murray Shanahan, Christos Kaplanis, and Jovana Mitrovic. Encoders and ensembles for task-free continual learning. arXiv preprint arXiv:2105.13327, 2021.
>
> > Introduction to a new hyperparameter for each of the prediction ($\alpha$’s)
> $\to$ Even though we introduce a new hyperparameter, we empirically observe that the performance is not sensitive to the value of $\alpha$ (please refer to the table below). We train and evaluate our agent with widely used coefficients [1-4], $\alpha_a=\alpha_c={0.9, 0.99, 0.999}$, for moving average but we do not see significant performance differences among them.
>
> [1] Tarvainen et al., Mean teachers are better role models: Weight-averaged consistency targets improve semi-supervised deep learning results, NeurIPS 2017
> [2] Ramirez et al., Omega: Optimistic EMA Gradients, ICML 2023
> [3] Li et al., Boosting Certified Robustness with EMA Method and Ensemble Model, arxiv 2021
> [4] Pham et al., On the Pros and Cons of Momentum Encoder in Self-Supervised Visual Representation Learning, arxiv 2022
>
> **Table A. The Behavior-IL setup.**
> | CAMA | Val. Seen $SR_{last}\uparrow$ | Val. Seen $GC_{last}\uparrow$ | Val. Seen $SR_{avg}\uparrow$ | Val. Seen $GC_{avg}\uparrow$ | Val. Unseen $SR_{last}\uparrow$ | Val. Unseen $GC_{last}\uparrow$ | Val. Unseen $SR_{avg}\uparrow$ | Val. Unseen $GC_{avg}\uparrow$ |
> |----------------------|---------|---------|---------|----------|-----------|-------------|---------------|---------------|
> | $\alpha=0.999$ | $29.55 \pm 1.31$ | $39.42 \pm 2.50$ | $27.66 \pm 3.01$ | $36.41 \pm 2.31$ | $14.21 \pm 4.51$ | $27.97 \pm 4.24$ | $11.56 \pm 3.47$ | $24.11 \pm 1.25$ |
> | $\alpha=0.99$ | $31.38 \pm 1.23$ | $40.65 \pm 1.97$ | $29.68 \pm 1.70$ | $38.19 \pm 1.79$ | $14.13 \pm 2.21$ | $29.61 \pm 2.26$ | $12.67 \pm 3.13$ | $25.43 \pm 2.31$ |
> | $\alpha=0.9$ | $29.84 \pm 1.96$ | $40.45 \pm 1.93$ | $14.13 \pm 3.82$ | $28.71 \pm 3.38$ | $14.13 \pm 3.82$ | $28.71 \pm 3.38$ | $12.38 \pm 4.25$ | $24.99 \pm 1.12$ |
>
> **Table B. The Environment-IL setup.**
> | CAMA | Val. Seen $SR_{last}\uparrow$ | Val. Seen $GC_{last}\uparrow$ | Val. Seen $SR_{avg}\uparrow$ | Val. Seen $GC_{avg}\uparrow$ | Val. Unseen $SR_{last}\uparrow$ | Val. Unseen $GC_{last}\uparrow$ | Val. Unseen $SR_{avg}\uparrow$ | Val. Unseen $GC_{avg}\uparrow$ |
> |----------------------|---------|---------|---------|----------|-----------|-------------|---------------|---------------|
> | $\alpha=0.999$ | $28.62 \pm 2.11$ | $39.09 \pm 1.60$ | $37.78 \pm 0.52$ | $47.28 \pm 0.51$ | $13.41 \pm 1.16$ | $31.63 \pm 3.26$ | $14.16 \pm 2.94$ | $34.21 \pm 2.65$ |
> | $\alpha=0.99$ | $32.47 \pm 5.53$ | $41.22 \pm 3.68$ | $38.75 \pm 3.52$ | $47.77 \pm 3.24$ | $13.89 \pm 1.75$ | $31.17 \pm 1.81$ | $14.92 \pm 2.75$ | $34.18 \pm 2.15$ |
> | $\alpha=0.9$ | $30.11 \pm 3.31$ | $39.59 \pm 1.10$ | $37.26 \pm 3.02$ | $46.55 \pm 1.69$ | $13.98 \pm 2.88$ | $30.24 \pm 3.02$ | $14.40 \pm 3.75$ | $32.73 \pm 2.47$ |

---

> ### Author Response · Authors · 2023-11-22
> **Answers to the questions of Reviewer q27B (2/2)**
>
> > Another baseline I could think of is to use the loss as the weight to dynamically change the old logits. What are some intuitions on why confidence is better than loss to use to dynamically weight?
>
> $\to$ Interesting suggestion! While using loss values can potentially serve as a strong baseline as the reviewer mentioned, we choose to use confidence scores considering the aspects below.
>
> First, while a confidence score ranges from 0 to 1, the loss functions usually have unbounded ranges. For the weighted moving average, the weights are expected to be in the range from 0 to 1 and the sum of the weights becomes 1. Therefore, it is not suitable to use the unbounded loss value as the weight. As an alternative, we use the confidences which are bounded.
>
> Second, employing the loss may result in marginal updates to the stored logit, especially in the early stages of training. This leads to a slow improvement of teacher logits for distillation. On the other hand, using confidence enables quicker updates with well-learned logits, leading to a rapid update to more informative logits. This is because widely used loss functions adopt log-based functions of probability such as cross-entropy losses, while confidences are probability values themselves. Due to the logarithmic scale of loss functions, loss values obtained by low probabilities could be quite small, and thus, the moving average with such small values might lead to marginal updates on the logits. As we can see in Figure 3, the confidence of a stored logit and a newly encountered logit is near 0, *i.e.*, the prediction probability of a stored logit and the newly introduced logit is $p$ and $p+\epsilon$, respectively, where $p$ and $\epsilon$ is positive but near 0.  In this case,  a stored logit needs to be updated using a newly introduced logit (*i.e.*, logits obtained by a model that is slightly more trained) since a logit with a high probability of the ground truth label implies that it learned well about that class, as we mentioned above. However, if we use the loss as a criterion of CAMA, the stored logit will be marginally updated since ($log(p)$) becomes nearly infinite when $p$ is near 0 in  (*i.e.*, $\gamma \approx 0$). In contrast, since confidences are probability values themselves, they can be updated more quickly with better logits during the early stages of training, leading to a faster improvement in distillation performance.
>
> > equation (1), the $\gamma$ in front of $z$ is it missing the subscript $a$ and $c$? or is it a fixed value?
>
> $\to$ Yes, the subscripts of $a$ and $c$ were missing in $\gamma$ and we corrected them in the revision. Thank you for pointing this out.
>
> > for equation (2), how are the confidence scores calculated, mathematically
>
> $\to$ Let $s^a_{i,k}$ a $k^{th}$ recent confidence score of an action, $i$, for the ground truth class $i$. We compute the averaged confidence score of the class, $i$, as, $\bar{s}^a_i = \sum_{k=1}^N s^a_{i,k}$.
>
> Similarly, let $s^c_{j,l}$ a $l^{th}$ recent confidence score of an object class, $j$, for the ground truth and we compute the averaged confidence score of the object class, $j$, as, $\bar{s}^c_j = \sum_{l=1}^N s^c_{j,l}$.
>
> > I am not too familiar with the benchmark, but when is the object class used when making decisions? if you need to heat a mug, does your agent also need to recognize the object in front of it is a mug too?
>
> $\to$ Yes, the agent needs to recognize the object (*i.e.*, a mug) in front of it and predict the object class, “mug,” if the agent intends to interact (*e.g.*, pick up, put, *etc.*) with it.

---

> > ### Comment · Reviewer_q27B · 2023-11-22
> >
> > I appreciate the authors clearing up all my concerns, and I have updated my score accordingly.

---

### Official Review · Reviewer_z9k5 · 2023-10-27

**Soundness:** 3 good
**Presentation:** 3 good
**Contribution:** 2 fair
**Rating:** 6
**Confidence:** 4

**Summary:**

The paper studies the online continual learning problem without task boundary information in an imitation learning context. It adapts a distillation based approach that was developed for the setting with task boundaries. Prior methods store past model logits in memory and update these with a weighting scheme to prevent these logits of becoming outdated using task boundary information. Here the weights are determined dynamically via the confidence of the past N predictions of the model, not relying on task boundary information. The method shows improved task performance compared to a variety of continual learning baselines in a simulated household environment. The evaluation is split into a setting where the agents has to continually learn new behaviors and a setting where the agent has to continually adapt to new environments.

**Strengths:**

- The way the method and experimental setup is presented, it can be easily followed and understood.

- The method is tested against a variety of baselines representing different approaches to the continual learning problem.

- The method outperforms all baselines.

- Although the novelty is on the smaller side it becomes apparent how the work relates to prior approaches and what limitation is tackled. It is also shown that tackling exactly that limitation leads to better performance.

**Weaknesses:**

- Depending on the availability of compute it would be beneficial to increase the number of seeds (3->5) to reduce the standard errors of the results. At the moment the standard errors are often larger than the gaps in mean performance.

- Effect of the hyperparameter N (number of past confidences averaged over). I think it would be interesting and important to see how sensitive the method is to that hyperparameter. If it works only well for a specific N, a lot of hyperparameter tuning will be needed to find that N for a given environment. If performance is robust with respect to N it would make the method more easily applicable.

- The evaluation is split between environment changes and behavioral changes. I understand the point of separating these two factors, but why was the crossing of the factors not studied? What happens if two factors change at the same time?

- As shown in Figure 3, when a new tasks arrives confidence drops. However, there is also evidence that in OOD cases a model can be confidently wrong (Do Deep Generative Models Know What They Don't Know?, Nalisnick et. al.). How detrimental would that be for performance?

Formalities:

- Abstract, last sentence: "outperforms prior arts", was that supposed to be "outperforms prior state of the art" ?
- Figure 4: In the second paragraph of 5.3 that references the figure 4, "HEAT" is claimed to be the newly learned task and Pick2&Place the old one, while in the caption of the figure it is the other way around.

**Questions:**

- The performance of the "finetune" baseline with respect to the $SR_{last}$ and $GC_{last}$ metrics. If I understand the baseline correctly is that the currently trained policy is finetuned on the data of the next task without any memory. Should the performance on the final task not be . Why is the performance on the .

- Which episodes are stored in the memory M? Is it simply a queue? If yes then given that its capacity is such a small percentage of the total number of episodes (as said in the text that is the standard) and given the number of tasks for each of the evaluation episodes will most episodes not be about the same same task such that the logits there should be replaced with the current logits of the model?

- In the CAMA w/o D.C. baseline, was the fixed coefficient tuned for optimal performance of the baseline?

**Details Of Ethics Concerns:**

/

---

> ### Author Response · Authors · 2023-11-21
> **Answers to the questions of Reviewer z9k5 (1/3)**
>
> We thank reviewer z9k5 for the constructive feedback and the encouraging remarks on clear presentation and experiments with a variety of baselines with strong performance.
> We address your concerns as follows.
>
> > Depending on the availability of compute it would be beneficial to increase the number of seeds (3->5) to reduce the standard errors of the results. At the moment the standard errors are often larger than the gaps in mean performance.
>
> $\to$ As suggested, we are running our experiments in 2 more random seeds (*i.e.*, total of 5 different random seeds) for the Environment-IL setup, compare the best-performing baselines (*i.e.*, CAMA w/o D.C. and DER++), and add the result in Table 5 of the revision. With smaller standard errors, we observe that our CAMA still outperforms the baselines by noticeable margins. We will update the 5-seed results for the other baselines in the final manuscript.
>
> **Table A. The Environment-IL setup.**
> | Model | Val. Seen $SR_{last}\uparrow$ | Val. Seen $GC_{last}\uparrow$ | Val. Seen $SR_{avg}\uparrow$ | Val. Seen $GC_{avg}\uparrow$ | Val. Unseen $SR_{last}\uparrow$ | Val. Unseen $GC_{last}\uparrow$ | Val. Unseen $SR_{avg}\uparrow$ | Val. Unseen $GC_{avg}\uparrow$ |
> |----------------------|---------|---------|---------|----------|-----------|-------------|---------------|---------------|
> | CAMA | $29.42 \pm 0.34$ | $38.14 \pm 1.10$ | $34.87 \pm 2.46$ | $43.79 \pm 2.84$ | $14.37 \pm 0.47$ | $30.31 \pm 0.41$ | $15.90 \pm 0.95$ | $33.34 \pm 1.87$ |
> | CAMA w/o D.C. | $30.95 \pm 0.72$ | $38.36 \pm 0.51$ | $33.70 \pm 2.48$ | $42.48 \pm 2.34$ | $12.58 \pm 0.89$ | $28.24 \pm 0.63$ | $13.13 \pm 0.52$ | $29.44 \pm 0.90$ |
> | DER++ | $26.06 \pm 1.35$ | $35.83 \pm 0.60$ | $28.51 \pm 2.93$ | $37.66 \pm 3.63$ | $11.93 \pm 1.21$ | $28.63 \pm 1.51$ | $10.85 \pm 1.87$ | $26.38 \pm 2.21$ |
>
> > Effect of the hyperparameter N (number of past confidences averaged over). I think it would be interesting and important to see how sensitive the method is to that hyperparameter. If it works only well for a specific N, a lot of hyperparameter tuning will be needed to find that N for a given environment. If performance is robust with respect to N it would make the method more easily applicable.
>
> $\to$ By sweeping $N=5, 10, 15, 20$, we evaluate our CAMA and summarize the results below. We do not observe significant performance differences, as observed in the tables, implying that our CAMA is not sensitive to the choice of the hyperparameter, $N$.
>
> It is noteworthy that in continual learning, it is less realistic to perform separate hyperparameter searches for each experimental setup since we do not know which data a model will encounter in the future. Thus, we conducted a hyperparameter search on the Behavior-IL setup and used the same hyperparameter for all setups including the  Environmental-IL and Behavior&Environment-IL setups (that we will introduce in the question below).
>
> **Table A. The Environment-IL setup.**
> | CAMA | Val. Seen $SR_{last}\uparrow$ | Val. Seen $GC_{last}\uparrow$ | Val. Seen $SR_{avg}\uparrow$ | Val. Seen $GC_{avg}\uparrow$ | Val. Unseen $SR_{last}\uparrow$ | Val. Unseen $GC_{last}\uparrow$ | Val. Unseen $SR_{avg}\uparrow$ | Val. Unseen $GC_{avg}\uparrow$ |
> |----------------------|---------|---------|---------|----------|-----------|-------------|---------------|---------------|
> | $N = 20$ | $31.92 \pm 2.83$ | $39.89 \pm 1.53$ | $38.51 \pm 1.17$ | $46.88 \pm 1.53$ | $12.74 \pm 2.16$ | $30.71 \pm 2.17$ | $13.67 \pm 1.86$ | $33.31 \pm 1.71$ |
> | $N = 15$ | $30.03 \pm 3.08$ | $38.94 \pm 3.37$ | $37.41 \pm 3.76$ | $46.94 \pm 3.20$ | $12.84 \pm 4.16$ | $29.60 \pm 2.52$ | $14.98 \pm 3.46$ | $33.07 \pm 3.20$ |
> | $N = 10$ | $32.47 \pm 5.53$ | $41.22 \pm 3.68$ | $38.75 \pm 3.52$ | $47.77 \pm 3.24$ | $13.89 \pm 1.75$ | $31.17 \pm 1.81$ | $14.92 \pm 2.75$ | $34.18 \pm 2.15$ |
> | $N = 5$ | $30.58 \pm 1.98$ | $39.92 \pm 1.13$ | $37.60 \pm 4.20$ | $46.89 \pm 4.04$ | $15.04 \pm 0.33$ | $30.53 \pm 0.13$ | $14.97 \pm 3.68$ | $33.65 \pm 3.20$ |

---

> > ### Comment · Reviewer_z9k5 · 2023-11-22
> >
> > Thank you for your detailed response to my concerns:
> >
> > - Seeds: I think the results are now more convincing.
> > - Sensitivity to N: The method seems robust to N. I think that is an added strength.
> > - Crossing of environments: Thank you, CAMA performs better in this setting as well.
> > - Overconfidence: That is true, I did not consider that.
> > - State of the art: I was not aware of that, thank you.
> > - Apologies for that comment, I think these were drafts of initial thoughts that I never formulated out properly and ended up dismissing.
> > - Memory: That clears it up, thanks!
> >
> > Given that my main concerns were all addressed I am going to increase my rating.

---

> ### Author Response · Authors · 2023-11-21
> **Answers to the questions of Reviewer z9k5 (2/3)**
>
> > The evaluation is split between environment changes and behavioral changes. I understand the point of separating these two factors, but why was the crossing of the factors not studied? What happens if two factors change at the same time?
>
> $\to$ Great suggestion! As suggested, we investigate the crossing of both Behavior-IL and Environment-IL setups, which we name the ‘Behavior&Environment-IL’ setup. We observe that our CAMA still outperforms the baselines by noticeable margins in the Behavior&Environment-IL setup, highlighting the efficacy of our proposed approach.
>
> **Table A. The Behavior&Environment-IL setup.**
> | Model | Val. Seen $SR_{last}\uparrow$ | Val. Seen $GC_{last}\uparrow$ | Val. Seen $SR_{avg}\uparrow$ | Val. Seen $GC_{avg}\uparrow$ | Val. Unseen $SR_{last}\uparrow$ | Val. Unseen $GC_{last}\uparrow$ | Val. Unseen $SR_{avg}\uparrow$ | Val. Unseen $GC_{avg}\uparrow$ |
> |----------------------|---------|---------|---------|----------|-----------|-------------|---------------|---------------|
> | EWC++ | $13.54 \pm 5.74$ | $19.30 \pm 4.61$ | $20.75 \pm 3.58$ | $28.41 \pm 3.12$ | $3.08 \pm 0.62$ | $13.44 \pm 1.55$ | $5.30 \pm 0.77$ | $20.94 \pm 2.16$ |
> | ER | $15.97 \pm 2.78$ | $22.07 \pm 2.30$ | $23.90 \pm 2.03$ | $31.48 \pm 1.76$ | $3.40 \pm 1.11$ | $13.30 \pm 0.73$ | $5.43 \pm 1.54$ | $19.25 \pm 2.24$ |
> | MIR | $13.19 \pm 0.92$ | $19.88 \pm 0.96$ | $24.45 \pm 2.35$ | $34.62 \pm 0.76$ | $2.47 \pm 0.62$ | $12.76 \pm 1.89$ | $6.02 \pm 1.55$ | $20.10 \pm 2.46$ |
> | DER++ | $14.23 \pm 0.92$ | $20.76 \pm 0.29$ | $21.32 \pm 1.92$ | $30.58 \pm 1.81$ | $8.02 \pm 0.82$ | $19.07 \pm 0.27$ | $5.56 \pm 1.26$ | $20.59 \pm 0.83$ |
> | Ours | $16.32 \pm 3.08$ | $22.80 \pm 2.57$ | $24.80 \pm 1.23$ | $33.03 \pm 2.14$ | $8.50 \pm 0.40$ | $20.30 \pm 1.92$ | $8.00 \pm 1.37$ | $21.49 \pm 2.08$ |
>
> > As shown in Figure 3, when a new task arrives confidence drops. However, there is also evidence that in OOD cases a model can be confidently wrong (Do Deep Generative Models Know What They Don’t Know?, Nalisnick et. al.) How detrimental would that be for performance?
>
> $\to$ The concern raised by Nalisnick et al. is about models exhibiting overconfidence in *incorrect classes*. We utilize confidence scores of only the *correct* classes (*i.e.*, ground-truth classes), thus there are no detrimental occurring due to the overconfidence problem.
>
> > Abstract, last sentence: “outperforms prior arts”, was that supposed to be “outperforms prior state of the art”?
>
> $\to$ Prior art and the state of the art are regarded as synonym (Please refer to the ‘Patent Law’ section of https://en.wikipedia.org/wiki/State_of_the_art). We will clarify it by “outperforms state of the arts.” Thank you!
>
> > Figure 4: In the second paragraph of 5.3 that references the figure 4, “HEAT” is claimed to be the newly learned task and Pick2&Place the old one, while in the caption of the figure it is the other way around.
>
> $\to$ Yes, you are correct. We are revising it in the revision. Thank you for pointing this out.
>
> > The performance of the “finetune” baseline with respect to the $SR_{last}$ and $GC_{last}$ metrics.
> >> If I understand the baseline correctly is that the currently trained policy is finetuned on the data of the next task without any memory.
>
> $\to$ Yes, as the reviewer mentioned, the “finetune” baseline is finetuned on the next task’s data without memory (*i.e.*, without any previous tasks’ data).
>
> >> Should the performance on the final task not be . Why is the performance of the .
>
> $\to$ We assume that the last part of the question is “Should the performance on the final task not be $SR_{avg}$. Why is the performance on the $SR_{avg}$.” (please correct us if we are wrong). The $SR_{avg}$ and $GC_{avg}$ metrics *do not* require the usage of memory. In other words, we measure metrics (*i.e.*, SR and GC) whenever right before receiving samples of a new behavior or environment and take the average of the metrics obtained at the end of each behavior or environment.
>
> > Which episodes are stored in the memory M? Is it simply a queue? If yes then given that its capacity is such a small percentage of the total number of episodes (as said in the text that is the standard) and given the number of tasks for each of the evaluation episodes will most episodes not be about the same task such that the logits there should be replaced with the current logits of the model?
>
> $\to$ The memory used is not a queue but an $M$-dimension array that does not have any temporal dependency on the order of incoming and outcoming examples. To address the concern that the current episodes may use the majority of memory space as the reviewer pointed out, we adopt reservoir sampling (Vitter et al., 1985, Rolnick et al., 2019, Buzzega et al., 2020) that enables us to maintain the stored episodes in memory approximately uniform across sequential tasks (*i.e.*, behaviors and environments).

---

> ### Author Response · Authors · 2023-11-21
> **Answers to the questions of Reviewer z9k5 (3/3)**
>
> > In the CAMA w/o D.C. baseline, was the fixed coefficient tuned for optimal performance of the baseline?
>
> $\to$ Yes, we tuned the coefficient to obtain the best valid unseen $SR_{last}$ for an arbitrarily chosen random seed in a setup (here, Environment-IL), among $\gamma={0.99, 0.9, 0.79, 0.5, 0.39, 0.01}$ and use it across all random seeds.
>
> To find the coefficient tuned for Environment-IL setup, we grid-search the best coefficients for the best performance (*i.e.*, the highest Val. Unseen $SR_{last}\uparrow$, following the same evaluation protocol of the ALFRED benchmark (Shridhar et al., 2021)), and summarize the results in Table A below. We observe that 'CAMA w/o D.C.’ achieves slightly better performance values than 'CAMA’ in some metrics (*e.g.*, Val. Unseen $SR_{last}\uparrow$). However, it underperforms CAMA in the other metrics by noticeable margins and requires additional searches for the coefficients for different setups.
>
> As we mentioned above, please note that it is often impractical to find the coefficients for the best performance considering all setups due to limited access to continuous data (*i.e.*, we do not know which data will come in). Thus, we used the same hyperparameter $\gamma$ for all setups including the  Environmental-IL and Behavior&Environment-IL setups.
>
> **Table A. The Environment-IL setup.** "CAMA w/o D.C." uses $\gamma_a = \gamma_c = 0.9$ found to yield the highest Val. Unseen $SR_{last}\uparrow$.
> | Model | Val. Seen $SR_{last}\uparrow$ | Val. Seen $GC_{last}\uparrow$ | Val. Seen $SR_{avg}\uparrow$ | Val. Seen $GC_{avg}\uparrow$ | Val. Unseen $SR_{last}\uparrow$ | Val. Unseen $GC_{last}\uparrow$ | Val. Unseen $SR_{avg}\uparrow$ | Val. Unseen $GC_{avg}\uparrow$ |
> |----------------------|---------|---------|---------|----------|-----------|-------------|---------------|---------------|
> | CAMA | $32.47 \pm 5.53$ | $41.22 \pm 3.68$ | $38.75 \pm 3.52$ | $47.77 \pm 3.24$ | $13.89 \pm 1.75$ | $31.17 \pm 1.81$ | $14.92 \pm 2.75$ | $34.18 \pm 2.15$ |
> | CAMA w/o D.C. | $30.27 \pm 1.61$ | $38.91 \pm 1.95$ | $38.44 \pm 3.89$ | $47.64 \pm 2.89$ | $14.08 \pm 1.32$ | $31.21 \pm 1.51$ | $14.31 \pm 2.89$ | $33.12 \pm 1.17$ |

---

> ### Author Response · Authors · 2023-11-22
> **Discussion reminder**
>
> We sincerely thank you for your effort in reviewing our submission. We gently remind the reviewer that we tried our best to address your concerns via our replies and revision of the manuscript. As the discussion period is nearing the end, we would be delighted to hear more from you if there are any further concerns.

---

### Official Review · Reviewer_UTbC · 2023-11-01

**Soundness:** 3 good
**Presentation:** 3 good
**Contribution:** 3 good
**Rating:** 6
**Confidence:** 4

**Summary:**

This work looks at the continuous learning problem for embodied agents that follow verbal task instructions. The outlined challenge is very relevant in the community as such agents need to constantly adapt to new tasks and environment challenges. In this work, the authors address Behavior-IL and Environment-IL through a newly proposed method, Confidence-Aware Moving Average, that updates an experience replay of stored logits based on model confidence. The proposed method is evaluated over a set of tasks and behaviors in the ALFRED simulation environment against state-of-the-art baselines.

**Strengths:**

* The proposed work addresses a relevant challenge for the deployment of embodied agents in challenging home environments, particularly addressing the problem of adapting previously learned policies to new environments and tasks.
* The idea of using confidence to update the stored logits is interesting, and potentially quite powerful (if there is any way to know that the confidence is reasonable and/or calibrated, see below).

**Weaknesses:**

* I think some of the assumptions this paper makes are not realistic. E.g., the paper indicates that this approach is task-free and that task-identifiers are not available, but then appears to assume that the data is balanced. In a real-world continual learning setup there is no guarantee that the tasks/data the agent encounters will be balanced, and so if data balance is enforced via sub-sampling I think this is implicitly leaking information regarding task identifiers (as they would be needed to balance an unbalanced dataset in practice). An analysis on how class/environment/behavior imbalance affects the model would be beneficial here.
* The performance improvements seem relatively marginal for many of the evaluation metrics. Were any statistical significance results run over these values?
* The training methodology is not entirely clear to me, and I think some of it should be pulled into the main paper as it reduces the clarity of the empirical results. Are all the models trained sequentially over a set of tasks/environments and then evaluated over the same set (in the seen case)? Do the seen/unseen evaluations use the same underlying trained model?

**Questions:**

1) Is there a concern about over/under-confidence with such an approach? Presumably, such a model is not calibrated since the model is continually out-of-distribution. Given that modern deep neural networks are particularly susceptible to confidence issues, an analysis on this aspect would be useful.
2) Does performance depend on the order of the tasks/environments? I see reference to random ordered sequences in Sec. B.3, but it's not clear to me how this ties into the training/evaluation methodology.
3) Is there any insight on how the size of the memory buffer relates to the number of learning tasks? Is 500 an arbitrary number or was this empirically chosen?

---

> ### Author Response · Authors · 2023-11-21
> **Answers to the questions of Reviewer UTbC (1/3)**
>
> We thank reviewer UTbC for the constructive feedback and the encouraging remarks on tackling a challenging problem and an interesting and powerful idea. We address your concerns as follows.
>
> > I think some of the assumptions this paper makes are not realistic. E.g., the paper indicates that this approach is task-free and that task-identifiers are not available, but it appears to assume that the data is balanced. But in a real-world continual learning setup, there is no guarantee that the tasks/data the agent encounters will be balanced, and so if data balance is enforced via sub-sampling I think this is implicitly leaking information regarding task identifiers (as they would be needed to balance an unbalanced dataset in practice).
>
> $\to$ For data balance, we follow common practice in previous continual learning literature (Wu et al., 2019, Aljundi et al., 2019, Bang et al., 2021, Boschini et al., 2022).  However, we agree with the reviewer in that we have no guarantee that the data that the agent encounters will be balanced. To investigate the imbalance scenario as suggested, we are performing additional experiments by removing the subsampling process in the Environment-IL setup and we will upload the result in the revision.
>
> Although we balance the training and evaluation episodes, we *do not* exploit any task-specific information or identifiers during the training and evaluation process, thus believe there is no information leakage. Since we denote by ‘task-free’ the absence of task identifiers *during the training process* (Aljundi et al., 2019), not task identifiers themselves. We added a discussion about task-free continual learning setups in “Task-free continual learning setups” in Sec. 2 of the revision.
>
> To consider a data-imbalance scenario, we consider *intra-task* imbalance, *i.e.*, an imbalance between object and action classes in each task to construct our proposed incremental setups. For *inter-task* balance, we followed previous works (Wu et al., 2019, Aljundi et al., 2019, Bang et al., 2021, Boschini et al., 2022) that assume balanced distribution between tasks (*i.e.*, assuming inter-task balance) for simplicity of setups. To mitigate severe inter-task imbalance in the environments in the ALFRED benchmark, we subsample the same number of episodes across all environments for inter-task balance.
>
> However, we agree that there is no guarantee of encountering inter-task-balanced data. To investigate our approach in such inter-task imbalance, we construct a modified version of the Environment-IL setup by omitting subsampling and summarize the result in Sec. E.1 of the revision. In this setup, an environment type (*i.e.*, Kitchens) has three times more episodes than the other environment types and thus, this setup considers inter-task imbalance as well.
>
> > An analysis on how class/environment/behavior imbalance affects the model would be beneficial here.
>
> $\to$ As the reviewer suggested, we are performing additional experiments on the Environment-IL setup with imbalances between environments (*i.e.*, without subsampling). We will upload the revision with the results soon.

---

> ### Author Response · Authors · 2023-11-21
> **Answers to the questions of Reviewer UTbC (2/3)**
>
> > The performance improvements seem relatively marginal for many of the evaluation metrics. Were any statistical significance results run over these values?
>
> $\to$ We also agree that the performance improvements seem relatively marginal, possibly due to large standard errors. Even though we increase the random seed, the magnitude of standard errors does not decrease well. We believe that this is because class distributions of task data are imbalanced in an incremental setup in embodied tasks, while balanced in existing class incremental setup in image classification tasks, and it amplifies the performance difference based on the order of tasks. More specifically, some tasks may share relatively many action and object classes, while others may share fewer classes. Previously learning such shared action and object classes may help better learning the current task (*i.e.*, forward transfer) and this implies that a model’s performance may depend on the order of the tasks (*i.e.*, how much the model learns the shared action and object classes ahead).
>
> For example, when learning to cool an object, acquiring knowledge of actions (such as opening/closing a fridge) and object classes (like apples, tomatoes, etc.) can also be used for subsequent learning to heat an object (e.g., heating an 'apple,' a 'tomato,' etc. by 'opening/closing' a microwave). We empirically observe that for the behavior, ‘Heat,’ our agent achieves 3.70% Valid Unseen SR after learning the behavior, ‘Cool,’ while it achieves zero Valid Unseen SR after learning the behavior, ‘Examine,’ which does have fewer shared object classes as illustrated in Figure 6 in Sec. E.2 in the appendix, implying the dependency of performance on a task order.
>
> While we agree that the performance improvements look marginal, we would like to note that a success rate (SR), the main metric that we use as much embodied AI work does, follows a *strict* evaluation protocol: a task is regarded as succeeded *only if* all the goal conditions are met and therefore, we may not observe significant improvements in the SR metric even if a model achieves some improvements in various aspects such as environmental representation [1] and language variation [2]. Sole comparison in this strict metric with significance test may pay less attention to these aspects, though they are valuable.
>
> [1] Gadre et al. Continuous scene representations for embodied AI. CVPR 2022.
>
> [2] Ishikawa & Sugiura. Moment-based Adversarial Training for Embodied Language Comprehension. ICPR, 2022.
>
>
> > The training methodology is not entirely clear to me, and I think some of it should be pulled into the main paper as it reduces the clarity of the empirical results.
>
> $\to$ We are sorry for the confusion. We address the questions below and add them in “Evaluation metrics” in Sec. 5 in the revision. Thank you for pointing this out.
>
> >> Are all the models trained sequentially over a set of tasks/environments and then evaluated over the same set (in the seen case)?
>
> $\to$ Partly, yes. All the models are trained sequentially over a set of tasks/environments but evaluated over *the episodes of the tasks/environments that the models have learned so far*. For example, if the model learned about the kitchen and bathroom, the evaluation set will only include data that is performed in the kitchen and bathroom (*i.e.*, not including the living room and the bedroom). For evaluation, we use episodes different from those used for training.
>
> >> Do the seen/unseen evaluations use the same underlying trained model?
>
> $\to$ Yes, we use the same trained model for both *seen* and *unseen* evaluations.

---

> ### Author Response · Authors · 2023-11-21
> **Answers to the questions of Reviewer UTbC (3/3)**
>
> > Is there a concern about over/under-confidence with such an approach? Presumably, such a model is not calibrated since the model is continually out-of-distribution. Given that modern deep neural networks are particularly susceptible to confidence issues, an analysis on this aspect would be useful.
>
> $\to$ To investigate the confidence issue that the reviewer mentioned, we calibrate the confidence scores of our CAMA using temperature scaling [1] and summarize the results below. As shown in the table, we do not observe significant performance differences, implying that the confidence issues may not be a primary concern in the proposed approach.
>
> [1] Guo et al. On calibration of modern neural networks. ICML, 2017.
>
> **Table A. The Environment-IL setup.**
> | Model | Val. Seen $SR_{last}\uparrow$ | Val. Seen $GC_{last}\uparrow$ | Val. Seen $SR_{avg}\uparrow$ | Val. Seen $GC_{avg}\uparrow$ | Val. Unseen $SR_{last}\uparrow$ | Val. Unseen $GC_{last}\uparrow$ | Val. Unseen $SR_{avg}\uparrow$ | Val. Unseen $GC_{avg}\uparrow$ |
> |----------------------|---------|---------|---------|----------|-----------|-------------|---------------|---------------|
> | CAMA | $32.47 \pm 5.53$ | $41.22 \pm 3.68$ | $38.75 \pm 3.52$ | $47.77 \pm 3.24$ | $13.89 \pm 1.75$ | $31.17 \pm 1.81$ | $14.92 \pm 2.75$ | $34.18 \pm 2.15$ |
> | CAMA w/ calibrated | $29.80 \pm 1.90$ | $38.73 \pm 1.63$ | $37.70 \pm 3.39$ | $47.44 \pm 2.71$ | $13.83 \pm 3.66$ | $30.49 \pm 0.63$ | $13.36 \pm 2.54$ | $31.85 \pm 2.27$ |
>
> > Does performance depend on the order of the tasks/environments? I see reference to random ordered sequences in Sec. B.3, but it’s not clear to me how this ties into the training/evaluation methodology.
>
> $\to$ Yes, performance depends on the order of the tasks and this is also widely observed in continual learning literature (Prabhu et al., 2020, bang et al., 2021, koh et al., 2022, ). To avoid favoring a particular order of the behaviors and environments, we validate our method with multiple random orders and report their mean and standard error ( sentence in Sec. 3.2.1).
>
> For the training/evaluation methodology, we use the same training and evaluation method across the entire randomly ordered sequences (*i.e.*, it does not require any additional tuning for a specific order), which enables our method to be applied to arbitrary sequences of the behaviors and environments.
>
>
> > Is there any insight on how the size of the memory buffer relates to the number of learning tasks? Is 500 an arbitrary number or was this empirical chosen?
>
> $\to$ We set the memory size to *less than 5%* of the training dataset following a common practice on continual learning literature (Prabhu et al., 2020; Bang et al., 2021; Koh et al., 2022; 2023). Specifically, we set our memory size to 500, which is approximately 2.38% of the training episodes in the ALFRED benchmark.

---

> ### Comment · Reviewer_UTbC · 2023-11-22
>
> Thanks for the response, I appreciate the effort required to run additional experiments during the rebuttal.
>
> Regarding class imbalance, did you have results for an updated imbalanced dataset? All I see in Sec. E1 is a class frequency histogram.
>
> I also want to push back against the assertion that prior works assume balanced datasets. In fact, many prior works explicitly do *not* assume balanced datasets as it is not uncommon in practice and can cause catastrophic forgetting in continual learning settings, e.g. Aljundi et al 2019 and Bang et al 2019 (only minor classes are balanced here).

---

> > ### Author Response · Authors · 2023-11-23
> > **Discussion reminder (closing in a few hours)**
> >
> > We sincerely appreciate your effort in reviewing our submission. We gently remind the reviewer that we tried our best to address your concerns via our replies and manuscript revision. As the discussion period is nearing its end (closing in a few hours), we would be glad to hear more from you if there are further concerns.

---

> ### Author Response · Authors · 2023-11-22
> **Additional response by the authors to the reviewer UTbC's response**
>
> > Regarding class imbalance, did you have results for an updated imbalance dataset? All I see in Sec. E1 is a class frequency histogram.
>
> $\to$ Yes, we have the quantitative result for an updated imbalance dataset in the table below, as we promised in our previous answer. We construct the imbalance dataset by removing the subsampling process used in the Environment-IL setup, resulting in imbalanced numbers of episodes across all environments. We observe that even with such an imbalance, our CAMA still outperforms the best-performing baseline (DER++) by noticeable margins. In particular, we observe significant improvements of $SR_{avg}$ and $GC_{avg}$ in both valid seen and unseen splits. We summarize the results in Table 3 in Sec.D.6 of the revision. We will add the results of the other baselines in the final manuscript.
>
> **Table A. The imbalanced Environment-IL setup.**
> | Model | Val. Seen $SR_{last}\uparrow$ | Val. Seen $GC_{last}\uparrow$ | Val. Seen $SR_{avg}\uparrow$ | Val. Seen $GC_{avg}\uparrow$ | Val. Unseen $SR_{last}\uparrow$ | Val. Unseen $GC_{last}\uparrow$ | Val. Unseen $SR_{avg}\uparrow$ | Val. Unseen $GC_{avg}\uparrow$ |
> |----------------------|---------|---------|---------|----------|-----------|-------------|---------------|---------------|
> | DER++ | $32.93 \pm 1.32$ | $45.16 \pm 0.94$ | $31.97 \pm 2.41$ | $43.56 \pm 2.95$ | $13.25 \pm 0.32$ | $29.05 \pm 0.31$ | $11.55 \pm 1.76$ | $28.75 \pm 1.46$ |
> | Ours | $34.72 \pm 0.98$ | $46.19 \pm 1.41$ | $39.92 \pm 2.55$ | $49.90 \pm 2.72$ | $14.63 \pm 0.70$ | $28.84 \pm 1.37$ | $18.67 \pm 1.88$ | $34.61 \pm 1.17$ |
>
> > I also want to push back against the assertion that prior works assume balanced datasets. In fact, many prior works explicitly do *not* assume balanced datasets as it is not uncommon in practice and can cause catastrophic forgetting in continual learning settings, e.g. Aljundi et al 2019 and Bang et al 2019 (only minor classes are balanced here).
>
> $\to$ Our argument about data balance with the references (Aljundi et al, 2019 and Bang et al, 2019) is that the number of data for each task is equal though the data distribution among classes within a task is imbalanced, *e.g.*, each task has 10,000 instances in 5 split CIFAR-10 and 5 split CIFAR-100, resulting in the number of task data being balanced between all tasks. Sorry for the confusion.

---

### Official Review · Reviewer_oYjZ · 2023-11-05

**Soundness:** 3 good
**Presentation:** 3 good
**Contribution:** 3 good
**Rating:** 6
**Confidence:** 3

**Summary:**

This paper studies continual learning for instruction-following embodied agents. The authors argue that continual learning is a more realistic setting for embodied AI. To evaluate embodied agents in a continual learning setting, the authors first propose two new continual learning setups for embodied agents:  Behavior Incremental Learning (Behavior-IL) and Environment Incremental Learning (Environment-IL) for new behavior and new environment learning, separately. Then, the authors proposed a new approach for weighting the new logits and old logits. Specifically, a confidence-aware moving average approach which updates logits based on confidence scores of class labels is proposed. The experimental results show that the proposed approach outperforms competitive baselines on the two proposed setups.

**Strengths:**

Originality and Significance:
The paper is well-motivated. Continual learning for embodied agents is a more realistic yet less explored area in our community. In addition, the two new continual learning setup seems reasonable and could be valuable for evaluating continual learning agents.


Quality:
The paper is technically sound. Most of the claims are supported by experimental results.


Clarity:
The paper is generally well-organized and implementation details are presented. However, some technical details may need further clarification. Please see the Weakness section for more details.

**Weaknesses:**

1. While the empirical results seem promising, the reviewer is not fully convinced by the proposed confident-aware moving average approach. Specifically, it is unclear why the average of most recent N confidence scores could be a good indicator of the ‘quality’ of the newly obtained logits. More concretely, the behavior $t_i$ of the current episode is different from the behavior $t_j$ that is associated with the sampled past experience $x’$. Why would confidence scores of current episodes with behavior $t_i$ provide information on a different behavior $t_j$.


2. In equation (1), subscript $a, c$ are missing in $\gamma_a$ and $\gamma_c$

**Questions:**

1. Please elaborate why the confidence scores are good indicators of the ‘quality’ of the new logits. Giving simple examples that demonstrate the effectiveness could be very helpful.

2. This paper uses imitation learning to train the agents. Is the proposed approach extendable to a reinforcement learning setting?

---

> ### Author Response · Authors · 2023-11-21
> **Answers to the questions of Reviewer oYjZ (1/1)**
>
> We thank the reviewer oYjZ for the constructive feedback and the encouraging remarks on a well-motivated, reasonable, and valuable idea, supporting experiments, and clear presentation. We address your concerns as follows.
>
> > While the empirical results seem promising, the reviewer is not fully convinced by the proposed confident-aware moving average approach.
>
> >> Please elaborate why the confidence scores are good indicators of the ‘quality’ of the new logits. Giving simple examples that demonstrate the effectiveness could be very helpful.
>
> $\to$ We argue that the confidence score partially estimates how well the model has learned about respective classes by which we define the `quality’ of the logit, similar to previous work measures confidence as a metric for the informativeness of stored prior models for distillation (*i.e.*, high confidence of the past model implies that there is much information to distill from the previous model) (Koh et al., 2023).
>
> For example, If the model $p$ predicts $p(i) = 1$ for the class $i$, it implies that the model has learned the class $i$ well, *i.e.*, it may contain ample information about the class $i$. Conversely, if the model predicts $p(i) = 0$, it implies that the model has yet learned the class $i$, *i.e.*, it may contain little information about the class $i$. We add this discussion in Sec. E.1 in the appendix in the revision.
>
> >> The behavior $t_i$ of the current episode is different from the behavior $t_j$ that is associated with the sampled past experience $x’$. Why would confidence scores of current episodes with behavior provide information on a different behavior $t_j$. .
>
> $\to$ We believe that the confidence scores of the current episodes with a behavior, $t_i$, can provide information on a different behavior, $t_j$, because the actions (*e.g.*, turn left, pick up, *etc.*) and object classes (*e.g.*, apple, pen, *etc.*) learned for the current task (*i.e.*, behavior and environment) can be also used for the previous tasks and therefore, achieving high accuracy of the actions and object classes for the current task can positively affect the previous tasks. For example, an agent that previously learned how to heat a mug may now learn to move two mugs on the table for the current task. To complete the current task, the agent should first learn to plan actions (*e.g.*, turn left/right, move ahead, pick up, *etc.*) and predict relevant object classes (*e.g.*, mug) to reach and interact with a mug. It is noteworthy that these learned actions and object classes are also used for the previous task (*i.e.*, heat a mug) as the agent should also reach and interact with a mug to heat it.
>
> > In equation (1), subscripts $a$, $c$ are missing in $\gamma_a$ and $\gamma_c$.
>
> $\to$ Thank you for the comment!. We revised the equation (1) ($\gamma$ -> $\gamma_a$, $\gamma$ -> $\gamma_c$) in the revision.
>
> > This paper uses imitation learning to train the agents. Is the proposed approach extendable to a reinforcement learning setting?
>
> $\to$ Great suggestion! Yes, our proposed approach can be extendable to a reinforcement learning setting. Once an action logit, $z_{old,a}'$, is stored in episodic memory for the current state, $s$, and action, $a$, the next state, $s’$, and the corresponding reward, $r$, resulting in the form of $(s, a, s’, r)$, we can maintain $(s, a, s’, r, z_{old,a}')$ and given a new logit, $z_a'$, for the same current state, we can update the previous logit, $z_{old,a}'$, in the same way as CAMA, resulting in $(s, a, s’, r, z_{new,a}')$ where $z_{new,a}'$$ = (1-\gamma_a) z_{old,a}' + \gamma_a z_a'(z_{new,c}'$ can be updated in the same way if applicable). Note that we do not assume the usage of imitation learning for training.

---

> ### Author Response · Authors · 2023-11-22
> **Discussion reminder**
>
> We sincerely thank you for your effort in reviewing our submission. We gently remind the reviewer that we tried our best to address your concerns via our replies and revision of the manuscript. As the discussion period is nearing the end, we would be delighted to hear more from you if there are any further concerns.

---

> ### Author Response · Authors · 2023-11-23
> **Discussion reminder (closing in a few hours)**
>
> We sincerely appreciate your effort in reviewing our submission. We gently remind the reviewer that we tried our best to address your concerns via our replies and manuscript revision. As the discussion period is nearing its end (closing in a few hours), we would be glad to hear more from you if there are further concerns.

---

### Author Response · Authors · 2023-11-21
**General response**

We thank the reviewers for their helpful feedback and encouraging comments including well-motivated idea (**oYjZ, q27B**), challenging setups (**UTbC**), reasonable idea (**oYjZ**), valuable and powerful approach (**oYjZ, UTbC**), effective approach (**q27B**), supporting experiments (**oYjZ, z9k5, q27B**), and clear presentation (**oYjZ, z9k5, q27B**).

We have uploaded the first revision of the manuscript (changes are highlighted by red color).

We will also upload the answer for reviewer q27B soon.

---

### Author Response · Authors · 2023-11-22
**Second revision**

We have uploaded the second revision of the manuscript. The revision includes additional experiments with a new setup, to address the reviewers’ concerns and suggestions (**UTbC**).

Summary of the changes
 - Add comparison with state of the art in the Environment-IL setup in an imbalanced scenario (**UTbC**).

---

### Meta-Review · Area_Chair_KDrD · 2023-12-17

**Metareview:**

This work accurately points out that embodied agents are actually in a continual learning setup and so should continue to update their models throughout their experience rather than the independent sample settings traditionally used.  They introduce two setups for learning what to do and where to act.  Their method weighting relies on a confidence aware moving average. There are a number of challenges to demonstrating the effectiveness of the approach which include the more severe class imbalances and difficulty in ALFRED and the need to construct fair evaluation splits.

**Justification For Why Not Higher Score:**

The results demonstrate some nice effectiveness and room for exploration but are not at this point strong enough or general enough to know if research on other EAI tasks should adopt the proposed methodology.

**Justification For Why Not Lower Score:**

The paper is clearly written/explained, the motivation relevant, and the results promising.

---

### Decision · Program_Chairs · 2024-01-16

Accept (poster)